# A Wave-Targeted Essentially Non-Oscillatory 3D Shock-Capturing Scheme for Breaking Wave Simulation

**Giovanni Cannata *** , **Federica Palleschi** , **Benedetta Iele** and **Francesco Gallerano**

Department of Civil, Constructional and Environmental Engineering, Sapienza University of Rome, 00184 Rome, Italy; federica.palleschi@uniroma1.it (F.P.); benedetta.iele@uniroma1.it (B.I.); francesco.gallerano@uniroma1.it (F.G.)
* Correspondence: giovanni.cannata@uniroma1.it

**Abstract:** A new three-dimensional high-order shock-capturing model for the numerical simulation of breaking waves is proposed. The proposed model is based on an integral contravariant form of the Navier–Stokes equations in a time-dependent generalized curvilinear coordinate system. Such an integral contravariant form of the equations of motion is numerically integrated by a new conservative numerical scheme that is based on three elements of originality: the time evolution of the state of the system is carried out using a predictor–corrector method in which exclusively the conserved variables are used; the point values of the conserved variables on the cell face of the computational grid are obtained using an original high-order reconstruction procedure called a wave-targeted essentially non-oscillatory scheme; the time evolution of the discontinuity on the cell faces is calculated using an exact Riemann solver. The proposed model is validated by numerically reproducing several experimental tests of breaking waves on computational grids that are significantly coarser than those used in the literature to validate the existing 3D shock-capturing models. The results obtained with the proposed model are also compared with those obtained with a previously published model, which is based on second-order total variation diminishing reconstructions and an approximate Riemann solver usually adopted in the existing 3D shock-capturing models. Through the above comparison, the main drawbacks of the existing 3D shock-capturing models and the ability of the proposed model to simulate breaking waves and wave-induced currents are shown. The proposed 3D model is able to correctly simulate the wave height increase in the shoaling zone and to effectively predict the location of the wave breaking point, the maximum wave height, and the wave height decay in the surf zone. The validated model is applied to the simulation of the interaction between breaking waves and an emerged breakwater. The numerical results show that the proposed model is able to simulate both the large-scale circulation patterns downstream of the barrier and the onset of quasi-periodic vortex structures close to the edge of the barrier.

**Keywords:** three-dimensional model; wave breaking; conservative scheme; high-order reconstructions; exact Riemann solver; contravariant formulation

## 1. Introduction

In the three-dimensional numerical simulation of wave propagation from the deep-water region to the coastline (including the surf zone), the main difficulty is related to numerical simulation of wave breaking. The three-dimensional models proposed by [1–4] are based on the idea, adopted in the Boussinesq-type models by Tonelli et al. [5,6], Roeber et al. [7], and Gallerano et al. [8], according to which a breaking wave can be simulated as a shock wave, i.e., as a discontinuity of the solution of the equations of motion. It is known that the numerical simulation of shock waves presents considerable difficulties. The numerical approximation of a shock wave may have the wrong strength (i.e., erroneous ratio between the value of the quantities before and after the shock wave) and an erroneous propagation speed, and thus it can mistake the position of the discontinuity at a given

time. Furthermore, during the simulation, spurious unphysical oscillations can arise in the vicinity of the discontinuities and can propagate on the rest of the solution, compromising the entire result. In the numerical simulation of hydrodynamic fields, the use of a conservative scheme is a necessary but not a sufficient condition for the convergence to the weak solution with discontinuity (of the equations of motion): conservative schemes must be able to preserve, at a discrete level, the properties of the continuum system, such as the conservation of global quantities and the conservation of invariants across a discontinuity. As underlined by Toro [9], in the numerical integration of differential equations of motion, the convective terms written in a form other than the divergence form do not allow the formulation of conservative schemes. Non-conservative numerical schemes do not guarantee the convergence to the correct weak solution with discontinuity. Furthermore, conservative numerical schemes (in which convective terms are written in a divergence form) applied to equations of motion expressed in terms of primitive variables ($H$ and $\vec{u}$, where $H$ is the water depth and $\vec{u}$ is the Cartesian flow velocity vector) can produce shock waves with erroneous propagation speeds. The integral form of the equations of motion expressed in terms of conserved variables ($H$ and $H\vec{u}$) allows for the formulation of conservative numerical schemes that can converge to the correct weak solution with discontinuity.

In [1–4], to simulate breaking waves as shock waves, the three-dimensional equations of motion are numerically integrated by shock-capturing finite volume numerical schemes that adopt second-order total variation diminishing (TVD) reconstruction techniques and approximate Riemann solvers. As demonstrated in the present paper, the simulation of breaking waves carried out by the above-mentioned numerical scheme are affected by some drawback: they underestimate the wave height increase during the shoaling process; they are not able to find the correct initial wave breaking point and are not able to correctly simulate the wave height decrease in the surf zone, with consequent underestimation of the wave-induced currents. In order to provide more reliable representations of the wave height evolution, wave breaking dynamics in the surf zone, and wave-induced coastal currents, such low-order schemes require the use of fine grids (especially in the horizontal directions) that limit their application mainly to laboratory-scale case studies.

In this paper, we demonstrate that the main cause of these deficiencies in the numerical results is related to two drawbacks of the above-mentioned numerical schemes. The first one is the fact that in those schemes the time evolution of the state of the system is obtained by updating the primitive variables $H$ and $\vec{u}$ (instead of the conserved variables $H$ and $H\vec{u}$) that, as written before, can produce shock waves with erroneous propagation speed. The second drawback is that, to limit the spurious oscillations that can appear in the vicinity of the discontinuities and avoid the propagation of such disturbances in the numerical solution, all the above-mentioned three-dimensional numerical models [1–4] adopt shock-capturing numerical schemes based on low-order (not more than second-order) TVD reconstruction techniques and approximate Riemann solvers.

In this paper, to overcome the drawbacks of the above-mentioned three-dimensional numerical models, the integral contravariant form of the Navier–Stokes equations proposed by Cannata et al. [3] is used to realize a new high-order shock-capturing numerical scheme for the three-dimensional simulation of the wave motion and wave-induced currents. The main elements of novelty of the proposed numerical scheme are three.

The first one is an original conservative numerical scheme in which the time evolution of the state of the system is carried out by updating the conserved variables $H$ and $H\vec{u}$. The second element of novelty is the proposal of an original high-order technique for the reconstruction of the point values of the conserved variables at the center of the computational cell faces. This original technique is specifically designed for the three-dimensional simulation of breaking waves and is called (in this paper) a wave-targeted essentially non-oscillatory (WTENO) technique. The third element of novelty consists in the fact that, differently from the models proposed in [1–4] (in which an approximate Riemann

solver is used), in this paper, we propose an exact Riemann solver for the time advancing of the point values of the conserved variables at the cell faces.

The paper is organized in the following form. In Section 2 the mathematical formulation and the proposed original numerical scheme based on conserved variables are presented. In the first subsection of Section 3, we describe the original high-order WTENO reconstruction technique adopted in the proposed shock-capturing scheme. In the second subsection of Section 3, the Riemann problem is presented. In Section 4, we validate the proposed model against several experimental tests and compare the numerical results also with those obtained using an alternative scheme based on weighted essentially non-oscillatory (WENO) reconstructions and the model proposed by Cannata et al. [3], which is based on second-order TVD reconstructions and an approximate Riemann solver usually adopted in the literature. In the last subsection of Section 4, we present a real application of the proposed model, consisting in the simulation of the complex three-dimensional flow velocity fields and free-surface elevation generated by the interaction between breaking waves and an emerged barrier. In the last section, the conclusions are drawn. In Appendix A, the strategy for the exact solution of the $x^1$-split Riemann problem is shown. In Appendix B, the symbols used in this paper are listed.

## 2. Mathematical Formulation

In the proposed three-dimensional numerical model, the motion equations are the integral contravariant incompressible Navier–Stokes equations written in a time-dependent curvilinear coordinate system proposed by Cannata et al. [3]. These equations are obtained by expressing the mass conservation and the momentum balance for an incompressible fluid on a moving control volume.

We denote by $\Delta V(t)$ the moving control volume and by $\Delta A(t)$ its boundary surface. In a Cartesian coordinate system, the integral form of the mass conservation equation on the moving control volume reads

$$\frac{d}{dt} \int_{\Delta V(t)} dV + \int_{\Delta A(t)} \left( \vec{u} - \vec{\omega} \right) \cdot \vec{n} \, dA = 0 \tag{1}$$

where $\vec{u}$ is the Cartesian flow velocity vector; $\vec{\omega}$ is the velocity vector of the boundary-surface control volume; $\vec{n}$ is the outward-normal unit vector, and the dot indicates the dot product between vectors (and between vectors and second-order tensors).

On the same moving control volume, the projection of the integral form of the momentum balance equation in the direction defined by a generic vector $\vec{\lambda}$ reads

$$\frac{d}{dt} \int_{\Delta V(t)} \vec{\lambda} \cdot \vec{u} \, dV \quad + \int_{\Delta A(t)} \vec{\lambda} \cdot \left( \vec{u} \otimes \left( \vec{u} - \vec{\omega} \right) \cdot \vec{n} \right) dA$$
$$- \int_{\Delta A(t)} \left( G\eta + \frac{p}{\rho} \right) \vec{\lambda} \cdot \left( \underline{I} \cdot \vec{n} \right) dA - \int_{\Delta A(t)} \frac{1}{\rho} \vec{\lambda} \cdot \left( \underline{R} \cdot \vec{n} \right) dA = 0 \tag{2}$$

where $\otimes$ indicates the tensor product between vectors; $G$ is the acceleration due to gravity; $\rho$ is the fluid density; $\eta = H - h$ is the free-surface elevation, in which $H$ and $h$ are, respectively, the total and still water depth; $p$ is the dynamic component of the total pressure, $P$, obtained by subtracting the hydrostatic component, $\rho G(\eta - x^3)$, from the $P$; $\underline{I}$ is the identity tensor; $\underline{R}$ is the stress tensor without the pressure term.

We denote by $x^i = x^i(\xi^1, \xi^2, \xi^3, \tau)$, $t = \tau$ and $\xi^i = \tilde{\xi}^i(x^1, x^2, x^3, t)$, $\tau = t$, respectively, a generic time-dependent coordinate transformation and its inverse, in which $(x^1, x^2, x^3, t)$ are the Cartesian coordinates and $(\xi^1, \xi^2, \xi^3, \tau)$ are the curvilinear ones. For this coordinate transformation, $\vec{g}_{(l)} = \partial \vec{x} / \partial \xi^l$ and $\vec{g}^{(l)} = \partial \xi^l / \partial \vec{x}$ $(l = 1, 3)$ are, respectively, the covariant and contravariant base vectors (in which letter $l$ between parentheses in the subscript or superscript indicates the $l - th$ base vector, not the $l - th$ component). We indicate by



$g^{m\alpha} = \overrightarrow{g}^{(m)} \cdot \overrightarrow{g}^{(\alpha)}$ the contravariant metric coefficients, and by $\sqrt{g} = \overrightarrow{g}_{(3)} \cdot \left( \overrightarrow{g}_{(1)} \wedge \overrightarrow{g}_{(2)} \right)$ the Jacobian of the transformation (in which $\wedge$ denotes the vector product).

We chose a particular time-dependent coordinate transformation, in which the horizontal coordinates, $\xi^1$ and $\xi^2$, are chosen to obtain a lateral boundary conforming grid, $\xi^1 = \xi^1(x^1, x^2, x^3)$, $\xi^2 = \xi^2(x^1, x^2, x^3)$, and the vertical coordinate, $\xi^3$, changes over time as a function of the water depth, $H$,

$$\xi^3 = \frac{x^3 + h(x^1, x^2)}{H(x^1, x^2, t)} \tag{3}$$

By Equation (3), the moving free surface is always identified by $\xi^3 = 1$ and the sea bottom by $\xi^3 = 0$. The coordinate transformation given by Equation (3) allows for expressing the Jacobian of the transformation in the form $\sqrt{g} = H\sqrt{g_0}$, in which $\sqrt{g_0} = \overrightarrow{k} \cdot \left( \overrightarrow{g}_{(1)} \wedge \overrightarrow{g}_{(2)} \right)$ and $\overrightarrow{k}$ is the vertical unit vector. We adopt a particular moving control volume whose boundaries are defined by surfaces on which only one curvilinear coordinate is constant. Thus, the infinitesimal moving control volume is expressed by $dV(t) = H(t)\sqrt{g_0}d\xi^1 d\xi^2 d\xi^3$ and the product between the outward-normal unit vector and the infinitesimal boundary surface on which $\xi^\alpha$ is constant is $\left( \overrightarrow{n} dA \right)|_{\xi^\alpha = const.} = \overrightarrow{g}^{(\alpha)} H\sqrt{g_0}d\xi^\beta d\xi^\gamma$ ($\alpha$, $\beta$, $\gamma$ are cyclic). Furthermore, we choose a particular vector $\overrightarrow{\lambda}$ for the projection of Equation (2): the vector $\overrightarrow{\lambda}$ is equal to the $l - th$ contravariant base vector defined at the center of the control volume, which is indicated by symbol $\widetilde{\overrightarrow{g}}^{(l)}$. The $m - th$ covariant components of vector $\widetilde{\overrightarrow{g}}^{(l)}$ is given by $\lambda_m = \widetilde{\overrightarrow{g}}^{(l)} \cdot \overrightarrow{g}_{(m)}$ and the contravariant components of the identity tensor are equal to the contravariant metric coefficients, $I^{m\alpha} = g^{m\alpha}$. Using the above-mentioned expressions and by adopting the Einstein notation (according to which an index repeated as subscript and superscript in a product represents summation over the range of the index), in the time-dependent curvilinear coordinate system, Equations (1) and (2) become

$$\frac{d}{dt} \int_{\Delta\xi^1 \Delta\xi^2 \Delta\xi^3} \left( H\sqrt{g_0} \right) d\xi^1 d\xi^2 d\xi^3$$
$$+ \sum_{\alpha=1}^{3} \left\{ \int_{\Delta A_0^{\alpha+}} \left( (Hu^\alpha - H\omega^\alpha)\sqrt{g_0} \right) d\xi^\beta d\xi^\gamma - \int_{\Delta A_0^{\alpha-}} \left( (Hu^\alpha - H\omega^\alpha)\sqrt{g_0} \right) d\xi^\beta d\xi^\gamma \right\} = 0 \tag{4}$$

$$\frac{d}{dt} \int_{\Delta\xi^1 \Delta\xi^2 \Delta\xi^3} \left( \lambda_m Hu^m \sqrt{g_0} \right) d\xi^1 d\xi^2 d\xi^3 +$$
$$\sum_{\alpha=1}^{3} \left\{ \int_{\Delta A_0^{\alpha+}} \lambda_m \left( Hu^m \left( \frac{Hu^\alpha}{H} - \omega^\alpha \right) + g^{m\alpha} \left( G\eta + \frac{p}{\rho} \right) H \right) \sqrt{g_0} d\xi^\beta d\xi^\gamma \right.$$
$$\left. - \int_{\Delta A_0^{\alpha-}} \lambda_m \left( Hu^m \left( \frac{Hu^\alpha}{H} - \omega^\alpha \right) + g^{m\alpha} \left( G\eta + \frac{p}{\rho} \right) H \right) \sqrt{g_0} d\xi^\beta d\xi^\gamma \right\} =$$
$$+ \sum_{\alpha=1}^{3} \left\{ \int_{\Delta A_0^{\alpha+}} \left( \lambda_m \frac{R^{m\alpha}}{\rho} H\sqrt{g_0} \right) d\xi^\beta d\xi^\gamma - \int_{\Delta A_0^{\alpha-}} \left( \lambda_m \frac{R^{m\alpha}}{\rho} H\sqrt{g_0} \right) d\xi^\beta d\xi^\gamma \right\} \tag{5}$$

where $\Delta\xi^1 \Delta\xi^2 \Delta\xi^3$ is the control volume in the transformed space, and $\Delta A_0^{\alpha+}$ and $\Delta A_0^{\alpha-}$ indicate, respectively, the control-volume boundary surfaces on which the coordinate $\xi^\alpha$ is constant, which are placed at larger and smaller values of $\xi^\alpha$ ($\alpha$, $\beta$, and $\gamma$ are cyclic). $R^{m\alpha}$ are the contravariant components of the stress tensor without the pressure term, $u^\alpha$ and $\omega^\alpha$ ($\alpha = 1, 3$) are, respectively, the contravariant component of the flow velocity and the velocity of the moving coordinate $\xi^\alpha$.

Equations (4) and (5) are the integral contravariant continuity and momentum balance equations expressed in terms of the conserved variables $H$ and $Hu^\alpha$. These equations can be discretized by a conservative finite-volume shock-capturing numerical scheme.

### 3. New 3D Shock-Capturing Numerical Scheme

*3.1. Finite-Volume Discretization*

We divide the physical domain occupied by the fluid into $n1 \times n2 \times n3$ hexahedral cells, $I_{i,j,k}$ ($i = 1, n1$; $j = 1, n2$; $k = 1, n3$), bounded by cell faces that lie on curvilinear coordinate surfaces (defined by indexes $i \pm \frac{1}{2}$, $j \pm \frac{1}{2}$ and $k \pm \frac{1}{2}$). We define the two-dimensional cell average of the water depth, $\left(\overline{H}\right)_{i,j} = 1/\left(\Delta\xi^1\Delta\xi^2\sqrt{g_0}\right)_{i,j} \int_{\left(\Delta\xi^1\Delta\xi^2\right)_{i,j}} H\sqrt{g_0}d\xi^1 d\xi^2$, and the three-dimensional cell average of the conserved variable, $\left(\overline{Hu^l}\right)_{i,j,k} = 1/\left(\Delta\xi^1\Delta\xi^2\Delta\xi^3\sqrt{g_0}\right)_{i,j,k}$ $\int_{\left(\Delta\xi^1\Delta\xi^2\Delta\xi^3\right)_{i,j,k}} Hu^l\sqrt{g_0}d\xi^1 d\xi^2 d\xi^3$. By integrating Equation (4) over the water column, and discretizing over space the obtained equation, we obtain the rate of change of the cell-averaged water depth,

$$
\begin{aligned}
\left(\overline{H}\right)_{i,j} = - \quad & \frac{1}{\left(\Delta\xi^1\Delta\xi^2\sqrt{g_0}\right)_{i,j}} \sum_{k=1}^{n3} \Big\{ \Big[ \left(Hu^1\sqrt{g_0}\Delta\xi^2\Delta\xi^3\right)_{i+\frac{1}{2},j,k} \\
& - \left(Hu^1\sqrt{g_0}\Delta\xi^2\Delta\xi^3\right)_{i-\frac{1}{2},j,k} \Big] \\
& + \Big[ \left(Hu^2\sqrt{g_0}\Delta\xi^1\Delta\xi^3\right)_{i,j+\frac{1}{2},k} - \left(Hu^2\sqrt{g_0}\Delta\xi^1\Delta\xi^3\right)_{i,j-\frac{1}{2},k} \Big] \Big\}
\end{aligned}
\tag{6}
$$

By discretizing over space Equation (5), we obtain the rate of change of the cell-averaged conserved variable, $\left(\overline{Hu^l}\right)_{i,j,k}$:

$$
\begin{aligned}
\frac{\partial\left(\overline{Hu^l}\right)_{i,j,k}}{\partial t} = - \quad & \left(\frac{\lambda_m}{\Delta\xi^1\Delta\xi^2\Delta\xi^3\sqrt{g_0}}\right)_{i,j,k} \Big\{ \Big[ \Big(Hu^m\frac{Hu^1}{H} \\
& - \left(G\eta + \frac{p}{\rho}\right)Hg^{m1}\Big)\sqrt{g_0}\Delta\xi^2\Delta\xi^3 \Big]_{i+\frac{1}{2},j,k} \\
& - \Big[ \Big(Hu^m\frac{Hu^1}{H} - \left(G\eta + \frac{p}{\rho}\right)Hg^{m1}\Big)\sqrt{g_0}\Delta\xi^2\Delta\xi^3 \Big]_{i-\frac{1}{2},j,k} \\
& + \Big[ \Big(Hu^m\frac{Hu^2}{H} - \left(G\eta + \frac{p}{\rho}\right)Hg^{m2}\Big)\sqrt{g_0}\Delta\xi^1\Delta\xi^3 \Big]_{i,j+\frac{1}{2},k} \\
& - \Big[ \Big(Hu^m\frac{Hu^2}{H} - \left(G\eta + \frac{p}{\rho}\right)Hg^{m2}\Big)\sqrt{g_0}\Delta\xi^1\Delta\xi^3 \Big]_{i,j-\frac{1}{2},k} \\
& + \Big[ \Big(Hu^m\left(\frac{Hu^3}{H} - \omega^3\right) - \left(G\eta + \frac{p}{\rho}\right)Hg^{m3}\Big)\sqrt{g_0}\Delta\xi^1\Delta\xi^2 \Big]_{i,j,k+\frac{1}{2}} \\
& - \Big[ \Big(Hu^m\left(\frac{Hu^3}{H} - \omega^3\right) - \left(G\eta + \frac{p}{\rho}\right)Hg^{m3}\Big)\sqrt{g_0}\Delta\xi^1\Delta\xi^2 \Big]_{i,j,k-\frac{1}{2}} \\
& + \frac{1}{\rho}\Big( \left[R^{m1}H\sqrt{g_0}\Delta\xi^2\Delta\xi^3\right]_{i+\frac{1}{2},jk} - \left[R^{m1}H\sqrt{g_0}\Delta\xi^2\Delta\xi^3\right]_{i-\frac{1}{2},j,k} \\
& + \left[R^{m2}H\sqrt{g_0}\Delta\xi^1\Delta\xi^3\right]_{i+\frac{1}{2},jk} - \left[R^{m2}H\sqrt{g_0}\Delta\xi^1\Delta\xi^3\right]_{i-\frac{1}{2},jk} \\
& + \left[R^{m3}H\sqrt{g_0}\Delta\xi^1\Delta\xi^2\right]_{i+\frac{1}{2},jk} - \left[R^{m3}H\sqrt{g_0}\Delta\xi^1\Delta\xi^2\right]_{i-\frac{1}{2},jk} \Big) \Big\}
\end{aligned}
\tag{7}
$$

*3.2. Time Advancing of the Numerical Solution*

*Step 1*

At the generic time $t$, the point values of the conserved variables at the cell faces are reconstructed, starting from their cell-average values, using an original procedure (described in Section 3.3) that is based on a high-order combination of polynomials centered in the computational cell. After the reconstruction procedure, on the cell face defined by indexes $i + \frac{1}{2}, j, k$, two different point values of the generic conserved variables are obtained, $f^-_{i+\frac{1}{2},j,k}$ and $f^+_{i+\frac{1}{2},j,k}$; the first one indicates the reconstruction obtained by the combination of polynomials centered in cell $I_{i,j,k}$; the second one indicates the reconstruction obtained by the combination of polynomials centered in cell $I_{i+1,j,k}$ (analogously on the other cell faces). At the end of this step, a couple of point values of the conserved variables on each cell face are obtained:

$$\left[H^-, \left(Hu^l\right)^-; H^+, \left(Hu^l\right)^+\right]_{i+\frac{1}{2},j,k}, \left[H^-, \left(Hu^l\right)^-; H^+, \left(Hu^l\right)^+\right]_{i,j+\frac{1}{2},k}, \left[\left(Hu^l\right)^-; \left(Hu^l\right)^+\right]_{i,j,k+\frac{1}{2}} \tag{8}$$

*Step 2*

At every cell face, the point values of the conserved variables obtained in the previous step are used as initial values of a local Riemann problem in which two constant states are separated by a discontinuity. The local Riemann problem is solved in exact form. For this purpose, at the center of every cell face, a transformation of the basis vectors is carried out, to obtain a local system in which one of the basis vectors is normal to the cell face and the other two lie on the plane of the cell face. By expressing the point values of the vector components with respect to the above local system of basis vectors, the local Riemann problem at each cell face can be solved as a Cartesian Riemann problem. The exact solution of the local Riemann problem provides the updated point values of the conserved variables on the cell faces. After a new basis vector transformation, from the local Cartesian one to the contravariant one, we obtain the contravariant components of the updated point values of the conserved variables on every cell face:

$$\left[H^{RS}, \left(Hu^l\right)^{RS}\right]_{i+\frac{1}{2},j,k}, \left[H^{RS}, \left(Hu^l\right)^{RS}\right]_{i,j+\frac{1}{2},k}, \left[\left(Hu^l\right)^{RS}\right]_{i,j,k+\frac{1}{2}} \tag{9}$$

where the superscript *RS* indicates point values obtained using the exact Riemann solver.

*Step 3*

The point values obtained in the previous step are introduced in the discretized form of the momentum balance equation (Equation (7)) in which the dynamic component of the pressure is omitted. The resulting equation provides an approximate field, $\left(\overline{Hu^l}\right)^*$, (called predictor field and denoted by the asterisk) of the cell average of the conserved variables $Hu^l$. This predictor field $\left(\overline{Hu^l}\right)^*$ represents an approximation of the final vector field $Hu^l$, since it is obtained without taking into account the dynamic component of the pressure.

*Step 4*

Differently from what is done in [3], in the proposed conservative scheme the predictor field $\left(Hu^l\right)^*$ is directly used to define the known right-hand side of a Poisson-like equation,

$$\frac{\partial\left[g^{ls}\frac{\partial\Phi}{\partial s}H\sqrt{g_0}\right]}{\partial\xi^l} = -\frac{\partial\left(Hu^l\right)^*\sqrt{g_0}}{\partial\xi^l} \tag{10}$$

in which $\Phi$ is an unknown potential scalar function. It must be noted that the right-hand side of Equation (10) is expressed directly in terms of the conserved variables $\left(Hu^l\right)^*$. Consequently, the scalar field of $\Phi$ is calculated by numerically solving Equation (10), without the need of calculating an approximate field of the water depth, $H^*$, and primitive variables, $u^{l*}$. Equation (10) is discretized by the same finite volume–finite difference method proposed by [3] and is numerically solved for $\Phi$ by an iterative method in which a four-color zebra, a line-by-line Gauss–Seidel procedure, and a multigrid technique are adopted.

*Step 5*

The gradient of the potential scalar $\Phi$ is used to correct the field of the conserved variables, both at the center of the computational cells and at the center of the cell faces:

$$\left(Hu^l\right) = \left(Hu^l\right)^* + Hg^{ls}\frac{\partial\Phi}{\partial s} \tag{11}$$

The final field of the conserved variables, $\overline{Hu^l}$, takes into account the dynamic component of the pressure and is associated with a divergence-free flow velocity field (unless

the approximation error by which the Poisson equation is numerically solved). By this predictor–corrector procedure, the proposed finite volume conserved numerical scheme makes use exclusively of conserved variables and, consequently, can converge to the weak solution with discontinuity, with a correct shock-speed propagation.

*Step 6*

The corrected values of the conserved variables, $Hu^l$, at the center of the vertical cell faces (defined by indexes $i \pm \frac{1}{2}, j, k$ and $i, j \pm \frac{1}{2}, k$) are associated with a divergence-free flow velocity field) and are introduced in the discretized integral over the water column of the continuity equation (Equation (6), to update the cell average of the water depth $\overline{H}$.

*Step 7*

The new position of the free surface is used to update the coordinates of all the computational cells, the metric terms associated to the time-dependent curvilinear coordinate system, and the contravariant velocity component of the moving curvilinear vertical coordinate, which is given by

$$\omega^3 = -\frac{\partial \xi^3}{\partial t} \tag{12}$$

*3.3. The Proposed Fifth-Order Wave-Targeted Essentially Non-Oscillatory Reconstruction Technique*

In this section, we propose an original high-order technique for the reconstruction of the point values of the conserved variables at the center of the cell faces. The high-order shock-capturing numerical scheme proposed in this paper is based on the idea that a breaking wave can be represented as a discontinuity of the numerical solution. It is known [9] that in the numerical solution with discontinuities, spurious numerical oscillation can arise close to the discontinuities and can propagate on the solution, compromising the entire result. As emphasized by Toro [9], the numerical schemes that ensure a non-oscillatory behavior are first-order accurate.

The shock-capturing numerical schemes proposed by [1–4] for the three-dimensional simulation of breaking waves are at most second-order accurate, where the solution is smooth and degrades to first-order accuracy near the discontinuities. The use of such low-order TVD schemes proposed in the literature is mainly due to the need to limit the spurious oscillation that can take place in the numerical solution.

As demonstrated in this paper, in the three-dimensional numerical simulation of breaking waves, second-order TVD shock-capturing schemes introduce too much numerical energy dissipation. The excessive numerical energy dissipation produces an underestimation of the wave height evolution in the shoaling phase, a wrong location of the initial wave breaking point, and an excessive reduction in the wave height in the surf zone. Further developments of second-order TVD methods are the WENO methods [10,11]. In the fifth-order upwind WENO schemes [11], to reconstruct the point values of a generic quantity in the face of a given computational cell $I_{i,j,k}$, three different second-order polynomials are defined, each one centered on a different cell in the neighborhood of $I_{i,j,k}$. In the above WENO scheme, the point value is obtained using a convex combination of the three polynomials. In such a convex combination, the initial weight of each polynomial is defined to obtain the maximum order of accuracy of the reconstruction. During the numerical simulation, the weight of each polynomial can be dynamically modified with respect to the original one, in order to reduce or increase its relative weight in the convex combination. The modification of the weights of the combination, with respect to the original values, produces a lower-order reconstruction that introduce numerical energy dissipation in the numerical solution. WENO schemes aim to maintain a high order of accuracy where the solution is smooth and to introduce numerical energy dissipation near the discontinuities, to limit the spurious oscillations. As shown in this paper, the use of a shock-capturing scheme based on fifth-order WENO reconstructions for the simulation of breaking waves, allows for only partially overcoming the drawbacks of shock-capturing schemes based on second-order TVD reconstructions. Numerical results of breaking waves carried out by a fifth-order

upwind WENO scheme are in better agreement with the experimental results (with respect to those obtained using a second-order TVD method), but show an underestimation of the wave height evolution in the shoaling zone, a wrong location of the initial breaking point, and an excessive reduction in the wave height in the surf zone. This is due to an excess of numerical energy dissipation introduced by the WENO scheme in the shoaling zone before the breaking point and in the entire wave breaking zone. Such excess numerical energy dissipation is due to the criterion by which the WENO schemes modify the weight of each polynomial of the convex combination. In the WENO schemes, the modification of the weights with respect to their original values is determined exclusively as a function of the smoothness of the polynomials. In the shoaling zone before wave breaking and in the entire surf zone, the steep wave fronts produce high gradients of the conserved variables ($H$ and $Hu^l$). In such a zone, in the presence of the above-mentioned high gradients, the WENO schemes consider the numerical solution as irregular and, consequently, modify the weights with respect to the original one. These modifications produce low-order reconstruction approximations that introduce too much numerical energy dissipation in the numerical solution.

It must be emphasized that the steepest fronts take place at the wave breaking, where the turbulence phenomena are more significant. At these steep fronts, the reconstruction procedure must introduce small numerical energy dissipation, to leave the task of dissipating the adequate amount of kinetic energy to the turbulence model. In the surf zone, the wave front is followed by a wave tail characterized by small energy dissipation due to turbulence. In such a portion of the wave, the reconstruction procedure must introduce the numerical energy dissipation required to contain the spurious oscillations that take place on the free surface at the steep wave fronts. In the present paper, we follow the conceptual line of the so-called targeted essentially non-oscillatory (TENO) schemes, first presented in [12] and improved by [13], and we propose an original high-order TENO scheme specifically designed for three-dimensional simulation of breaking waves, called wave-targeted essentially non-oscillatory (WTENO) scheme. The original WTENO scheme is based on three main elements. The first one is the definition of three different second-order polynomials, each one candidate to approximate the point value of the conserved variable on the faces of the generic computational cell $I_{i,j,k}$. The second main element is the definition of a regularity function, associated with each polynomial, which depends on the smoothness of the polynomials. The third main element of the proposed reconstruction technique is the definition of a dynamic threshold, common to the three polynomials, which varies as a function of both the smoothness of the polynomials and the steepness of the wave front. At every instant of the numerical simulation and at every point of the computational domain, the comparison between the regularity function and the dynamic threshold determines whether only one, two, or all the three candidate polynomials participate in the given reconstruction of the point values. In the case in which all the polynomials participate in the reconstruction, the maximum order of accuracy of the reconstruction is reached (fifth order), since the calculated point value is equivalent to the one obtained using a fourth-order polynomial defined on the stencil of five contiguous cells centered on the cell $I_{i,j,k}$. In the case in which only one or two polynomials participate in the reconstruction, the order of accuracy of the reconstruction reduces and the numerical energy dissipation introduced in the simulation by the numerical scheme increases. In the proposed WTENO scheme, the regularity function and the dynamic threshold are defined in such a way as to guarantee good non-oscillatory properties of the numerical scheme and avoid the excessive numerical energy dissipation produced by reconstruction techniques commonly used in the literature, such as TVD or WENO. The purpose of the proposed reconstruction procedure is to limit the numerical energy dissipation of energy introduced by the numerical scheme on the steep wave fronts (where the energy dissipation is left to the turbulence model) and ensure an adequate numerical energy dissipation on the non-breaking wave fronts and wave tails (where the energy dissipation due to the turbulence is lower and occurs mainly near the bottom).

In the three-dimensional finite volume proposed in this paper, the cell average value of the generic conserved variable is defined by the integral of the given variable over the volume of the computational cell. The reconstruction procedure consists in calculating the point values of each conserved variable on the center of every computational cell face, starting from the integral of the given conserved variable over the cell volume. In the proposed numerical model, we define a time-dependent coordinate transformation that allows us to transform the computational cells that in the physical space have an irregular and time-varying shape into computational cells that in the transformed space have a regular and fixed shape. In the transformed space, the integral of the conserved variables over the cell volume is defined by three consecutive one-dimensional integrals, each one defined on a single coordinate line. This makes it possible to define a three-dimensional reconstruction procedure for the calculation of the point values of the conserved variables that consists of three consecutive one-dimensional reconstructions, each one relative to a single coordinate line (dimension by dimension reconstructions).

In the first step of the procedure, which is carried out along coordinate $\xi^3$, the first level of reconstruction allows for passing from the values of the given variable averaged over the volume of the computational cell to the values of the same variable averaged over a surface on which only coordinates $\xi^1$ and $\xi^2$ vary. The values obtained in the first step are used as starting points of a second one-dimensional reconstruction, carried out along coordinate $\xi^2$. This reconstruction provides the values of the given variable averaged over lines on which only $\xi^1$ varies. The third one-dimensional reconstruction, which is carried out along coordinate $\xi^1$, allows for obtaining the point values of the given variable on the center of the cell face on which coordinate $\xi^1$ is constant.

The mathematical steps of the above procedure are shown below.

In the transformed space denoted by coordinates $\xi^1, \xi^2, \xi^3$, the generic computational cell $I_{i,j,k}$ has a regular and fixed shape and a volume equal to $\Delta\xi^1\Delta\xi^2\Delta\xi^3$. Because of the regularity of the computational cell in the transformed space, the cell average of the conserved variable $\overline{Hu^l}$ can be indicated by the following triple integral:

$$\overline{Hu^l} = \frac{1}{\Delta\xi^1\Delta\xi^2\Delta\xi^3} \int\limits_{\xi^3_{k-\frac{1}{2}}}^{\xi^3_{k+\frac{1}{2}}} \int\limits_{\xi^2_{j-\frac{1}{2}}}^{\xi^2_{j+\frac{1}{2}}} \int\limits_{\xi^1_{i-\frac{1}{2}}}^{\xi^1_{i+\frac{1}{2}}} Hu^l d\xi^1 d\xi^2 d\xi^3 \tag{13}$$

This integral is expressed as a sequence of three consecutive single integrals, each one defined along a different coordinate line. This implies that the reconstruction of the point values of $Hu^l$ can be calculated with three consecutive one-dimensional reconstructions, each one carried out along a different spatial coordinate [14].

We indicate by $(\overline{Hu_l})_{i,j,k}$ the average of the conserved variable $Hu^l$ over the cell $I_{i,j,k}$. We indicate by $\left(Hu^l\right)_{i+\frac{1}{2},j,k}$ and $\left(Hu^l\right)_{i-\frac{1}{2},j,k}$ the two point values of $Hu^l$ at the center of the cell faces on which $\xi^1$ is constant, which are placed at the side on which $\xi^1$ increases and decreases, respectively.

Step 1: starting from the cell averages $(\overline{Hu^l})_{i,j,k}$, along the coordinate $\xi^3$, we reconstruct the surface average $(\widetilde{Hu^l})_{i,j,k}$, which is defined by

$$(\widetilde{Hu^l})_{i,j,k} = \frac{1}{\Delta\xi^2} \int_{\xi^2_{j-\frac{1}{2}}}^{\xi^2_{j+\frac{1}{2}}} [\frac{1}{\Delta\xi^1} \int_{\xi^1_{i-\frac{1}{2}}}^{\xi^1_{i+\frac{1}{2}}} Hu^l\left(\xi^1,\xi^2,\xi^3\right) d\xi^1] d\xi^2 \tag{14}$$

Step 2: starting from the surface averages $(\widetilde{Hu^l})_{i,j,k}$, along the coordinate $\xi^2$, we reconstruct the line averages $(\widehat{Hu^l})_{i,j,k}$, which are defined by

$$(\widehat{Hu^l})_{i,j,k} = 1/\Delta\xi^1 \int_{\xi^1_{i-\frac{1}{2}}}^{\xi^1_{i+\frac{1}{2}}} Hu^l\left(\xi^1,\xi^2,\xi^3\right)d\xi^1 \tag{15}$$

Step 3: starting from the line averages $(\widehat{Hu^l})_{i,j,k}$, along the coordinate $\xi^1$, we reconstruct the point values on the cell faces $(Hu^l)_{i+\frac{1}{2},j,k}$ and $\left(Hu^l\right)_{i-\frac{1}{2},j,k}$.

Below, we show the original WTENO procedure by which, starting from the line averages $(\widehat{Hu^l})_{i,j,k}$, we reconstruct the point value $(Hu^l)_{i+\frac{1}{2},j,k}$. An analogous procedure is used for the reconstruction of the point values on the other cell faces.

The reconstruction of $(Hu^l)_{i+\frac{1}{2},j,k}$ is given by the value assumed at $\xi^1_{i+\frac{1}{2}}$ by a polynomial function $F_{i,j,k}(\xi^1)$ defined on cell $I_{i,j,k}$ (subscripts $i, j, k$ indicate the position of the computational cell):

$$\left(Hu^l\right)_{i+\frac{1}{2},j,k} = F_{i,j,k}\left(\xi^1_{i+\frac{1}{2}}\right) \tag{16}$$

Polynomial function $F_{i,j,k}(\xi^1)$ is given by a combination of three different second-order polynomials $P_{(p)i,j,k}(\xi^1)$ (with $p = -1, 0, 1$), each one defined on a sub-stencil of three contiguous cells $I_{i+p+q,j,k}$ (with fixed $p$ and $q = -1, 0, 1$):

$$P_{(p)i,j,k}\left(\xi^1\right) = a_{(p)i,j,k}\left(\xi^1\right)^2 + b_{(p)i,j,k}\left(\xi^1\right) + c_{(p)i,j,k} \tag{17}$$

For each polynomial, the coefficients $a_{(p)i,j,k}, b_{(p)i,j,k}, c_{(p)i,j,k}$ are uniquely determined by solving a linear system of three equations. The three equations of the system are defined by imposing that the integral of the polynomial over each cell of the stencil be equal to the line average of $Hu_l$ obtained at the end of the previous one-dimensional reconstruction:

$$(\widehat{Hu^l})_{i+p+q,j,k} = \frac{1}{\Delta\xi^1} \int_{\xi^1_{i+p+q-\frac{1}{2}}}^{\xi^1_{i+p+q+\frac{1}{2}}} P_{(p)i,j,k}\left(\xi^1\right)d\xi^1 \text{ (with fixed } p \text{ and } q = -1, 0, 1) \tag{18}$$

By using Equation (17) for $P_{(p)i,j,k}(\xi^1)$ and by analytically solving the integrals on the right-hand side of Equation (18), the coefficients of the polynomial are uniquely determined. The reconstruction of the point value on the cell face located at $\xi^1_{i+\frac{1}{2}}$ is given by a combination of the three third-order reconstructions, each one relative to a single polynomial:

$$F_{i,j,k}\left(\xi^1_{i+\frac{1}{2}}\right) = \Omega_{-1}P_{(-1)i,j,k}\left(\xi^1_{i+\frac{1}{2}}\right) + \Omega_0 P_{(o)i,j,k}\left(\xi^1_{i+\frac{1}{2}}\right) + \Omega_1 P_{(1)i,j,k}\left(\xi^1_{i+\frac{1}{2}}\right) \tag{19}$$

where $\Omega_p$ (with $p = -1, 0, 1$), which are called non-linear weights, are defined by

$$\Omega_p = \frac{\delta_p c_p}{\sum_{p=-1}^{1} \delta_p c_p} \tag{20}$$

in which $\delta_p$ are cut-off functions adopted in TENO schemes [12,13] that can be 0 or 1 and determine whether one, two, or all the three polynomials participate in the reconstruction. $c_p$ are the so-called linear weights that are defined so that the reconstruction of the point value of the given variable in $\xi^1_{i+\frac{1}{2}}$, which is obtained using all the three second-order polynomials, is equal to the reconstruction obtained using a single fourth-order polynomial defined on a big stencil composed by the five contiguous cells $I_{i+p+q,j,k}$ (with $p = -1, 0, 1$ and $q = -1, 0, 1$):

$$c_{-1} = 1/10; c_0 = 6/10; c_1 = 3/10 \tag{21}$$

These values guarantee that, in the case in which all the three polynomials participate in the reconstruction, the maximum order of accuracy (fifth order) is obtained. By following [13], in the proposed WTENO scheme the cut-off functions $\delta_p$ (with $p = -1, 0, 1$) are calculated by defining three regularity functions $\Gamma_p$ and a dynamic threshold $C_T$. At every instant of the simulation and at every point of the computational domain, the comparison between $\Gamma_p$ and $C_T$ determines whether one, two, or three candidate polynomials participate in the reconstruction:

$$\delta_p = \begin{cases} 0 & \Gamma_p < C_T \\ 1 & \Gamma_p \geq C_T \end{cases} \tag{22}$$

For the calculation of $\Gamma_p$, we adopt the same procedure of [13] and define a function $\gamma_p$ (relative to each polynomial $P_{(p)}$) that depends on the smoothness indicator $\beta_p$ of each polynomial and on the global smoothness indicator $\tau_p$:

$$\gamma_p = \left( C + \frac{\tau_p}{\beta_p + \epsilon} \right)^{\mu}, \text{ with } p = -1, 0, 1 \tag{23}$$

where $\epsilon = 1 \times 10^{-8}$ is a coefficient to avoid zero in the denominator and $\beta_p$ are computed by the expression proposed in [11] and usually adopted in WENO schemes. By following Borges et al. [15], in this paper the global smoothness indicator is defined as $\tau_p = |\beta_1 - \beta_{-1}|$. Coefficients $C$ and $\mu$ in Equation (23) are meant to improve the ability of the numerical procedure to detect the discontinuities and, at the same time, reduce the numerical energy dissipation introduced in the solution. In this paper $C$ and $\mu$ are set equal to 1 and 6, respectively, which are the values usually adopted in the TENO schemes [12,13]. Once calculated, function $\gamma_p$ is normalized to be between 0 and 1,

$$\Gamma_p = \frac{\gamma_p}{\sum_{p=-1}^{1} \gamma_p}, \text{ with } p = -1, 0, 1 \tag{24}$$

Using Equation (23), the values of $\Gamma_p$ are compared to the dynamic threshold, $C_T$, common to the three candidate polynomials. In the proposed WTENO scheme, differently from [13], the dynamic threshold $C_T$ varies with space and time not only as a function of the regularity of the polynomials but also as a function of the steepness of the wave front. For this purpose, we propose the following expression for $C_T$:

$$\begin{cases} C_T = 10^{-n} \\ n = B_l + (\theta + \theta_2)(B_h - B_l) \end{cases} \tag{25}$$

In Equation (25), $B_l$ and $B_h$ are integer parameters that determine the minimum and maximum values of exponent $n$. For all the numerical simulations carried out in this paper, we set $B_l = 1$ and $B_h = 7$. $\theta$ is a function proposed in [13] (ranging between 0 and 1) that depends exclusively on the smoothness indicator $\beta_p$ of each polynomial and the global smoothness indicator $\tau_p$. $\theta_2$ is a new function proposed in the present paper (equal to or greater than 0) that depends on the steepness of the wave front. Exponent $n$ can assume values equal to or greater than 1. For low values of $n$, the propensity of the proposed technique to cut off one or two candidate polynomials from the reconstruction increases. Function $\theta$ is calculated by the expression proposed by Peng et al. [13]:

$$\theta = \frac{1}{1 + \frac{1}{d} \max\left( \frac{\tau_p}{\beta_p + \epsilon} \right)}, \text{ with } p = -1, 0, 1 \tag{26}$$

where $d$ is a parameter set equal to 10. For values of $d$ larger than 10, the proposed technique is more dissipative, since it increases the propensity to cut off one or two candidate polynomials from the reconstruction. The new function $\theta_2$ is defined to guarantee a high

order of accuracy of the reconstructions on the steep wave fronts and limit the spurious oscillations that can arise on the wave tails,

$$\theta_2 = \left\{ \frac{\left( \frac{\partial \eta}{\partial t} - \Psi \right) + \left| \frac{\partial \eta}{\partial t} - \Psi \right|}{2 \left| \frac{\partial \eta}{\partial t} - \Psi \right|} \right\} \left[ \frac{\frac{\partial \eta}{\partial t}}{\Psi} - 1 \right] \tag{27}$$

in which $\eta$ is the free-surface elevation, $\partial \eta / \partial t$ is its local time rate of change, and $\Psi$ is the threshold value for $\partial \eta / \partial t$. In Equation (27), the quantity $\partial \eta / \partial t$ and its threshold value $\Psi$ are used to identify the breaking wave fronts and distinguish them from the wave tails. A gravity wave can be represented as a perturbation that, at a given point, produces temporal variations in the free-surface elevation. The portion of the wave in which $\partial \eta / \partial t$ is positive is associated with the wave front. Larger values of $\partial \eta / \partial t$ indicate a higher steepness of the wave front. The portion of the wave in which $\partial \eta / \partial t$ is negative is associated with the tail of the wave (the portion of the wave between a crest and the successive wave front). In the proposed reconstruction procedure, a wave is breaking if $\partial \eta / \partial t$ is higher than $0.3\sqrt{Gh}$, which is a threshold value in the typical range used for detecting the onset of wave breaking [16]. In Equation (27), the term in curly brackets is equal to 1 only at points of the domain and at instants in which $\partial \eta / \partial t$ is larger than its threshold value $\Psi$, and is equal to zero otherwise. This term has the task to activate function $\theta_2$ only at the breaking wave fronts, since it is null both at the tail of the wave and at non-breaking wave fronts. If different from zero, the magnitude of $\theta_2$ is determined by the value of the term in square brackets of Equation (27), which is greater than or equal to zero. This entails that the $\theta_2$ is higher the higher the steepness of the breaking wave front is: $\theta_2$ has a maximum at the beginning of the wave breaking (where the fronts are steeper) and progressively decreases as the wave breaking proceeds. As can be deduced from Equation (25), the higher the value of $\theta_2$, the lower the value of $C_T$. This entails that, at the wave breaking fronts, the propensity of the proposed technique to cut off one or two candidate polynomials is reduced. Consequently, at the wave breaking fronts, the introduction of numerical energy dissipation in the numerical solution due to the reconstruction procedure is lower.

At the tail of the waves and at the non-breaking fronts, where $\theta_2$ is equal to zero, the value of the dynamic threshold $C_T$ is a function of $\theta$, which depends exclusively on the regularity of the reconstruction polynomials. This entails that at the tail of the waves and at non-breaking fronts, the propensity of the proposed technique to cut off one or two candidate polynomials is higher. Consequently, in such a portion of the wave, the capacity of the proposed reconstruction procedure to introduce numerical energy dissipation in the solution is increased.

The reconstruction technique proposed in this section allows for limiting the energy dissipation introduced by low-order numerical schemes at the steep wave fronts (where the energy dissipation is entrusted mainly to the turbulence model) and guarantees an adequate numerical energy dissipation at the non-breaking wave fronts and at the tail of the waves (where the energy dissipation due to turbulence is lower and occurs mainly near the bottom).

### 3.4. The Solution of the Riemann Problem

The updating of the point values of the conserved variables at the center of the cell faces is carried out by solving an exact local Riemann problem. For this purpose, following the strategy proposed in [17] and adopted in [3], the vector of the conserved variables is expressed with respect to a local Cartesian system of base vectors. On this local system of base vectors, starting from the integral contravariant form of the Navier–Stokes equations adopted in this paper, by some mathematical manipulation we obtain a differential form of the equations of motion written in conservative form, in which the vertical coordinate varies over time according to the $\sigma$-coordinate transformation, while the horizontal coordinates

are equal to the Cartesian ones and the vector of the conserved variables is expressed with respect to the Cartesian basis vectors

$$\frac{\partial H}{\partial t} + \frac{\partial Hu}{\partial x^1} + \frac{\partial Hv}{\partial x^2} + \frac{\partial H(u^3 - \omega^3)}{\partial \xi^3} = 0 \tag{28}$$

$$\frac{\partial Hu}{\partial t} + \frac{\partial Huu}{\partial x^1} + \frac{\partial Huv}{\partial x^2} + 0.5\frac{\partial GH^2}{\partial x^1} - GH\frac{\partial h}{\partial x^1} + \frac{\partial Hu(u^3 - \omega^3)}{\partial \xi^3} = 0 \tag{29}$$

$$\frac{\partial Hu}{\partial t} + \frac{\partial Huu}{\partial x^1} + \frac{\partial Huv}{\partial x^2} + 0.5\frac{\partial GH^2}{\partial x^1} - GH\frac{\partial h}{\partial x^1} + \frac{\partial Hu(u^3 - \omega^3)}{\partial \xi^3} = 0 \tag{30}$$

$$\frac{\partial Hw}{\partial t} + \frac{\partial Huw}{\partial x^1} + \frac{\partial Hvw}{\partial x^2} + \frac{\partial Hw(u^3 - \omega^3)}{\partial \xi^3} = 0 \tag{31}$$

In Equations (28)–(31), $Hu, Hv, Hw$ are the local Cartesian components of the vector of the conserved variable, $x^1$ and $x^2$ are the horizontal Cartesian coordinates; $\xi^3$ is the moving curvilinear vertical coordinate; $u^3$ and $\omega^3$ are the contravariant components of the flow velocity and moving coordinate, respectively.

The above system can be rewritten in the following compact form:

$$\frac{\partial U}{\partial t} + \frac{\partial F(U)}{\partial x^1} + \frac{\partial G(U)}{\partial x^2} + \frac{\partial H(U)}{\partial \xi^3} = S \tag{32}$$

where the bold letters indicate the following vectors and matrices:

$$U = \begin{pmatrix} H \\ Hu \\ Hv \\ Hw \end{pmatrix}, F(U) = \begin{pmatrix} Hu \\ Huu + 0.5GH^2 \\ Huv \\ Huw \end{pmatrix}, G(U) = \begin{pmatrix} Hv \\ Huv \\ Hvv + 0.5GH^2 \\ Hvw \end{pmatrix}$$

$$H(U) = \begin{pmatrix} 0 \\ Hu(u^3 - \omega^3) \\ Hv(u^3 - \omega^3) \\ Hu(u^3 - \omega^3) \end{pmatrix}, S = \begin{pmatrix} -\frac{\partial H(u^3 - \omega^3)}{\partial \xi^3} \\ GH\frac{\partial h}{\partial x^1} \\ GH\frac{\partial h}{\partial x^2} \\ 0 \end{pmatrix} \tag{33}$$

By following the procedure proposed by Toro [9], the solution of the above system of equations is obtained by solving three different exact Riemann problems, each one relative to a coordinate direction. In Appendix A, the strategy for the solution of the exact $x^1$-split Riemann problem is shown.

## 4. Results and Discussion

In this section, the validation of the proposed numerical model and its real application to the simulation of the complex three-dimensional flow velocity fields generated by the interaction between breaking waves and an emerged barrier are presented. The model validation was carried out by numerically reproducing several experimental tests of breaking waves widely used in the literature. The same experimental tests were reproduced also using an alternative model based on fifth-order WENO reconstructions and the model proposed by Cannata et al. [3], which is based on second-order TVD reconstructions and an approximate Riemann solver.

Let us indicate with WTENO the numerical model proposed in this paper. The WTENO model adopts a reconstruction procedure based on the proposed original fifth-order WTENO reconstruction technique and the exact Riemann solver.

Let us indicate with 5WENO the model that differs from the proposed one only in the reconstruction procedure. The 5WENO model adopts a reconstruction procedure based on a fifth-order upwind weighted essentially non-oscillatory technique [11].

Let us indicate by 2TVD the numerical model proposed by Cannata et al. [3]. The 2TVD model adopts a reconstruction procedure based on a second-order TVD technique and an approximate (HLL) Riemann solver.

In all the above numerical models, the turbulent stress tensor is modeled by the Smagorinsky turbulence model, which is used in [1,3]. In the Smagorinsky model, the eddy viscosity $\nu_t$ is given by a simple algebraic expression:

$$\nu_t = (C_s \Delta)^2 \sqrt{2\underline{S} : \underline{S}} \tag{34}$$

where $\underline{S}$ is the strain rate tensor; the double dot indicates the doubly contracted product between second-order tensors; $\Delta = \left(\sqrt{g}\Delta\xi^1\Delta\xi^2\Delta\xi^3\right)^{1/3}$ is the filter width (that is related to the volume of the computational cells); and $C_s$ is the Smagorinsky coefficient, which is the only adjustable coefficient of the turbulence model. Although more sophisticated turbulence models could be used, in this paper we adopt this simple turbulence model to highlight the differences produced in the numerical solution by the different numerical models.

*4.1. Test 1: Spilling Breaking Wave*

For this subsection, an experimental test of monochromatic breaking waves carried out by Stive [18] and widely used to validate numerical models [3,5] was numerically reproduced using the proposed WTENO model, the above-described alternative numerical model (5WENO), and the model proposed by Cannata et al. [3] (2TVD). In the experimental test, regular waves with wave height and wave period equal to 0.158 m and 1.79 s, respectively, propagated along a 55 m-long channel, characterized by an initial constant water depth equal to di 0.85 m followed by a beach with a 1 : 40 slope (Figure 1). In the numerical simulations of this test, we adopted the same time and spatial discretization step for all the above-mentioned three numerical models. The time step was 0.001s. The number of grid points in the vertical direction was equal to 9. The spatial discretization along the wave propagation direction $\Delta x$ was set to obtain a given value of the ratio $r_L = n_x/L$, in which $n_x$ is the number of grid points along the wave propagation direction and $L$ is the wavelength in deep water. In the other horizontal direction, the spatial grid step $\Delta y$ was set equal to $2\Delta x$. For this test, the deep-water wavelength was $L = 4.2$m. In the three-dimensional numerical simulation of breaking waves, computational grids with $r_L < 100$ can be considered coarse grids. In the present paper, to highlight the drawbacks of the numerical models based on low-order reconstruction schemes, for this test we adopted a very coarse computational grid in which the discretization step in the wave propagation direction was $\Delta x = 0.075$ m, which corresponded to $r_L = n_x/L = 56$. In this test, the Smagorinsky coefficient $C_s$ was set to 0.06, for all three numerical models. Figure 1 shows an instantaneous free-surface elevation and velocity field obtained using the proposed WTENO model during the simulation of Test 1. From this figure, it is possible to see the steepening of the wave fronts approaching the coastline and the following wave height reduction due to the wave breaking. The contour of the eddy viscosity, $\nu_t$, shows that its maximum values are found at steep wave fronts of the breaking waves.

Figure 2 shows the comparison between the wave heights experimentally measured by Stive [18] and those obtained using the proposed WTENO model, the 2TVD model and the 5WENO model. From Figure 2, it can be seen that the results obtained with the proposed model show a general good agreement with the experimental results: the wave height increase in the shoaling zone was well simulated, although slightly overestimated, and the wave breaking point was correctly identified. From this figure it can be seen that by using a coarse grid ($r_L = 56$), the proposed model provides a good agreement with the experimental data: the wave height increase in the shoaling zone, the maximum wave height, and the wave height reduction in the surf zone were well simulated. Furthermore, the wave breaking point was correctly located by the proposed model. The same Figure shows that, on this very coarse grid, the 2TVD model provided significantly worse results

than those obtained using the proposed model: the increase in wave height before the wave breaking was not correctly simulated; the wave breaking point was located too far offshore and the maximum wave height was highly underestimated (with respect to the experimental data). From the same figure, it is possible to see that the results obtained with the 5WENO model, although better than those obtained with the 2TVD model, show an underestimation of the wave height in the shoaling zone and an incorrect location of the wave breaking point.

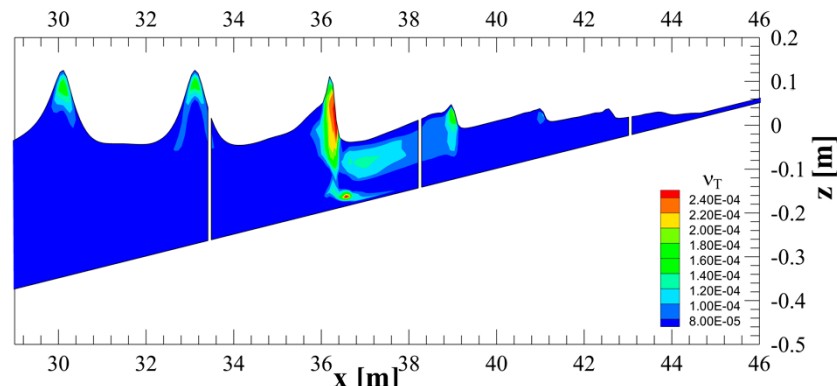

**Figure 1.** Test 1. Spilling breaking wave. Instantaneous free-surface elevation and contour of the eddy viscosity, $\nu_t$, obtained using the proposed WTENO model.

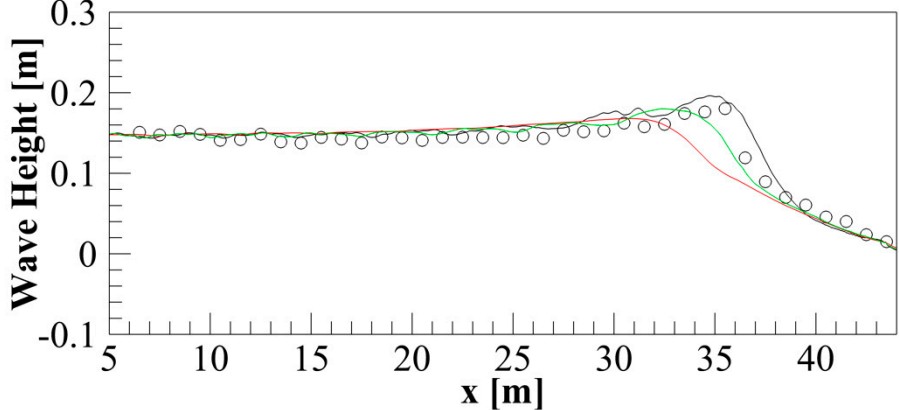

**Figure 2.** Test 1. Spilling breaking wave. Mean wave heights: comparison between experimental measurements by [18] (circles) and numerical results obtained with the WTENO model (black line), the 2TVD model proposed by Cannata et al. [3] (red line), and the 5WENO model (green line).

*4.2. Test 2: Spilling/Plunging Breaking Wave*

In this subsection, we test the ability of the proposed model to simulate waves characterized by a spilling/plunging breaker. This kind of wave represents the upper limit of the breaking waves that can be effectively simulated using the proposed numerical model. In fact, the proposed one belongs to the class of numerical models in which the position of the free surface is uniquely defined by a function of the horizontal coordinates. This class of numerical models cannot effectively simulate a fully plunging breaker, in which the crest of the wave curls over and drops onto the trough of the wave. For accurate numerical simulations of fluid flows with complex moving interfaces, more computationally expensive models based on a meshless Lagrangian approach [19,20] or finite element methods for two-fluid flows [21,22] are required. For this section, we numerically reproduced an experimental test of spilling/plunging breaker carried out by Stive [18] in the same experimental setup used for Test 1. The wave height was 0.142 m and the wave period was 2.99 s. The Iribarren number was $\xi_b = \tan(\alpha)/\sqrt{H_b/L} = 0.35$ (where $\alpha$ is the plain beach slope, $H_b$ is the wave

height at the breaking point, and $L$ is the deep-water wavelength), which was very close to the upper limit (0.4) beyond which the breaker is classified as plunging. As in Test 1, in this test the time step was 0.001 s and the number of grid points in the vertical direction was equal to 9. The spatial discretization along the wave propagation direction was the same used for the previous test, $\Delta x = 0.075$ m, and $\Delta y = 2\Delta x$. For this test, the deep-water wavelength was $L = 7.3$ m and the ratio between the number of grid points along the wave propagation direction, $n_x$, and $L$ was $r_L = 97$. This test was carried out also using the 2TVD model proposed by Cannata et al. [3] and the 5WENO model. For all the models, the Smagorinsky coefficient $C_s$ was set to 0.1. Figure 3 shows an instantaneous free-surface elevation and contour of the eddy viscosity obtained with the proposed WTENO model during the simulation of Test 2. From this figure, it is possible to see the increase in the wave height in the shoaling zone and the very steep front of the waves in the surf zone. As for the previous test, the maximum values of the eddy viscosity can be found at the very steep wave fronts that arise in the surf zone after the wave breaking point.

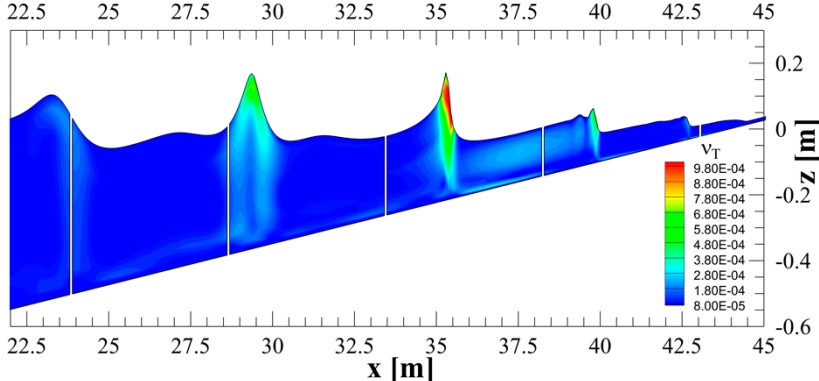

**Figure 3.** Test 2. Spilling/plunging breaking wave. Instantaneous free-surface elevation and contour of the eddy viscosity obtained using the proposed WTENO model.

Figure 4 shows the comparison between the wave height measured by Stive [18] and those obtained with the proposed WTENO model, the 2TVD model, and the 5WENO model. The proposed model provides results that are in quite good agreement with the experimental data: the wave height increase in the shoaling zone, the maximum wave height, and the wave breaking point were well predicted. The main discrepancies with the experimental data were found after the breaking point, in the first part of the surf zone, where the proposed model predicted slightly overestimated wave heights. This slight overestimation of the wave height in the surf zone can be attributed to the adopted turbulence model. For this simulation, the Smagorinsky turbulence model with a constant value of the Smagorinsky coefficient ($C_s = 0.1$) was not able to adequately represent the energy dissipation due to the wave breaking. In this test, rather than adjust the Smagorinsky coefficient in the surf zone (and increase the energy dissipation), we preferred to slightly underestimate the dissipation of energy introduced by the turbulence model, to highlight the differences produced by the three different numerical schemes. Figure 4 shows that the results obtained with the 2TVD model were considerably worse than those obtained with the proposed model: the wave height evolution in the shoaling was poorly predicted; the maximum wave height was significantly underestimated; the wave breaking point was located too far offshore, and the wave height reduction in the surf zone was incorrectly represented with respect to the experimental results. From the same figure, it can be noted that the results obtained using the 5WENO model are in better agreement with the experimental data, with respect to those obtained using the 2TVD model, but are affected by some drawbacks: the wave height increase in the shoaling zone was only partially well predicted; the maximum wave height was underestimated, and the wave breaking point was located slightly too far offshore.

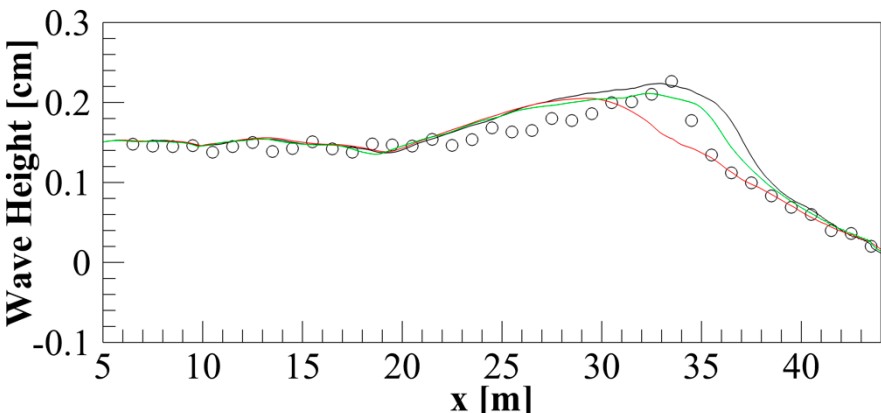

**Figure 4.** Test 2. Spilling/plunging breaking wave. Mean wave heights: comparison between experimental measurements by [18] (circles) and numerical results obtained with the WTENO model (black line), the 2TVD model proposed by Cannata et al. [3] (red line), and the 5WENO model (green line).

*4.3. Test 3: Cnoidal Wave*

In this subsection, we numerically reproduce an experimental test of a cnoidal wave proposed by Ting and Kirby [23] and used by several authors to validate shock-capturing models [1,4]. The experimental channel by [23] has an initial depth $h = 0.4$ m followed by a 1:35 sloping beach. In this test, a cnoidal wave with a wave height of $H = 0.125$ m and a wave period of $T = 2$ s was simulated. This experimental test was reproduced using the proposed WTENO model, the 2TVD model proposed by Cannata et al. [3], and the 5WENO model. As for the previous tests, in all the simulations shown in this section, the number of grid points in the vertical direction was equal to 9 and the time step was 0.001 s. For this wave, the deep-water wavelength was $L = 6.3$ m. It must be emphasized that in the literature this test is usually reproduced on very fine computational grids in which the spatial discretization step in the wave propagation direction is $\Delta x = 0.025$ m [1] and $\Delta x = 0.0375$ m [4], which corresponds to $r_L = n_x/L = 254$ and $r_L = 168$, respectively. For the proposed paper, we adopted the same coarse grid used for Test 1 and Test 2 ($\Delta x = 0.075$; $\Delta y = 2\Delta x$), which corresponds to $r_L = 84$. In this test, the Smagorinsky coefficient $C_s$ was set to 0.06, for all the three numerical models.

Figure 5 shows an instantaneous free-surface elevation and contour of the eddy viscosity obtained using the proposed WTENO model during the simulation of Test 3.

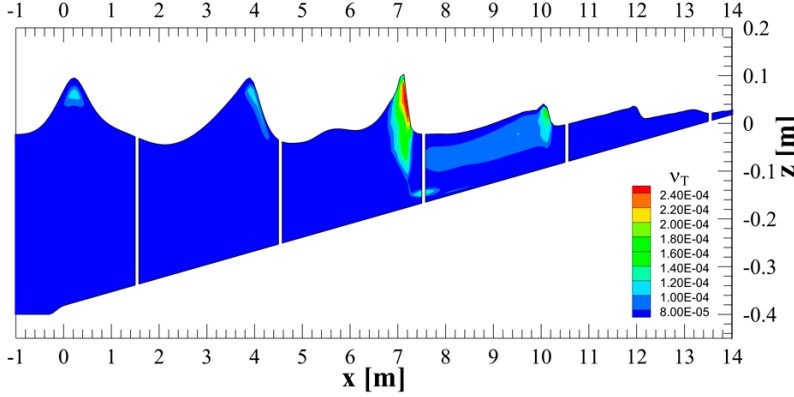

**Figure 5.** Test 3. Cnoidal wave. Instantaneous free-surface elevation and contour of the eddy viscosity obtained using the proposed WTENO model.

In Figure 6, the maximum, minimum, and average values of the free-surface elevation obtained using the proposed WTENO model are compared to the experimental data by [23]

and the numerical results obtained using the 2TVD model and the 5WENO model. The figure shows a good agreement between the results obtained with the proposed WTENO model and the experimental data: the wave height evolution in the shoaling zone was well predicted; the maximum wave height and the wave breaking point were correctly represented; a slight overestimation of wave height after the wave breaking point can be observed. This lack of energy dissipation in the first part of the surf zone can be attributed to the adopted turbulence model. In this test, as for the previous ones, we adopted a constant low value of the Smagorinsky coefficient ($C_s = 0.06$). With this choice, we limited the kinetic energy dissipation due to the turbulence model to highlight the effect produced by too dissipative numerical schemes on the wave height evolution. From Figure 6, it can be seen that, for this test, the 2TVD model provided very erroneous results: the wave height evolution in the shoaling zone was almost absent, the maximum wave height was highly underestimated, and the location of the wave breaking point was extremely incorrectly predicted. From the same figure, it can be noted that the results obtained using the 5WENO model are affected by drawbacks, that although lower, are similar to those of the 2TVD model: the wave height evolution in the shoaling zone and the maximum wave height were underestimated; the location of the wave breaking point was predicted too far offshore.

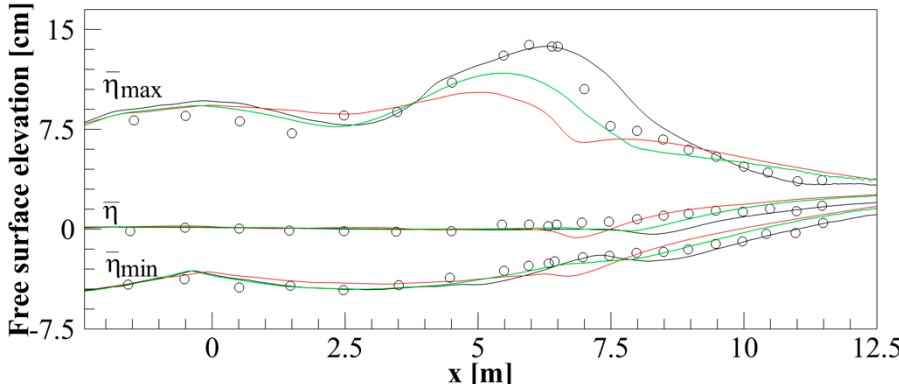

**Figure 6.** Test 3. Cnoidal wave. Maximum, minimum, and average values of the free-surface elevation. Comparison between experimental data by [23] (circles) and numerical results obtained using the proposed WTENO model (black line), the 2TVD model proposed by Cannata et al. [3] (red line), and the 5WENO model (green line).

From the comparison between the numerical results shown in Figures 2, 4 and 6, it can be deduced that the drawbacks in the results obtained by the models based on second-order TVD and fifth-order WENO reconstructions are mainly due to an excess of numerical energy dissipation introduced by the above models in the numerical solution. This excess of numerical energy dissipation is mainly due to the low order of the reconstruction procedure rather than to turbulence model that is a too dissipative. In fact, in the above simulations, the coefficient $C_s$ of the turbulence model (the Smagorinsky coefficient) was set to values (0.06 and 0.1, respectively), which are lower than or equal to the minimum value (0.1) usually adopted in the literature. From the same figures it can be noted that, although the numerical results obtained using the 5WENO model are in better agreement with the experimental data than those obtained using the 2TVD model, the WENO reconstructions introduced too much numerical energy dissipation in the shoaling and surf zone compared to the proposed WTENO technique. The reason for such excessive numerical energy dissipation can be found in the criterion by which the WENO technique determines the weight of each reconstruction polynomial. In the WENO technique, the weight of each candidate polynomial is determined exclusively as a function of the smoothness of the polynomials. In the shoaling zone before the wave breaking and in the whole surf zone, the steep wave fronts produce high gradients of the conserved variables ($H$ and $Hu_l$). In such steep wave fronts, the WENO technique interprets the numerical solution as irregular and,

consequently, modifies the original weight of each polynomial to provide a lower weight to the more irregular ones. This entails a final low-order point value reconstruction that is significantly more dissipative than the original fifth-order one. It must be emphasized that these too dissipative WENO reconstructions occur in the same computational cells where the maximum kinetic energy dissipation due to the turbulence model takes place (i.e., on the steepest wave fronts). The main consequence is an excess of kinetic energy dissipation introduced in the numerical solution by the reconstruction technique, which produces underestimated wave heights in the shoaling zone, an erroneous wave breaking location, and a maximum wave height that is too low.

By the same figures, it is possible to deduce that the above-mentioned drawbacks of the WENO technique are overcome by the WTENO reconstruction procedure proposed in this paper. In fact, on the steep wave fronts, where the maximum energy dissipation due to the turbulence model takes place, the dynamic threshold, defined in the proposed WTENO scheme to determine whether one or two polynomials must be excluded from the reconstruction procedure, assumes its maximum values. Consequently, on these steep wave fronts, all the three candidate polynomials participate in the reconstruction with weights that are not modified with respect to the original ones, so that the point values of the conserved variables are given by a fifth-order low-dissipative reconstruction. This entails that, on the steep wave fronts, no further numerical energy dissipation is introduced by the reconstruction procedure, in addition to the one introduced by the turbulence model. For this reason, in the simulations obtained using the WTENO model, the wave height in the shoaling and surf zone was significantly higher than that obtained using the 5WENO model. Inside the shoaling and surf zone, the waves propagating towards the coastline exhibited a steep wave front followed by a wave tail. On these wave tails, where the local time derivative of the free-surface elevation was negative, the dynamic threshold defined in the proposed WTENO scheme depended only on the regularity of the candidate polynomials. This implies that, on this portion of the waves, the magnitude of the dynamic threshold can be reduced and the contribution of the more irregular polynomials can be excluded from the procedure, producing a low-order reconstruction that introduces numerical energy dissipation in the numerical solution. Therefore, this numerical energy dissipation is introduced mainly on the wave tails, where the energy dissipation due to the turbulence model is lower. In this way, it is possible to obtain good non-oscillatory properties of the numerical model without excessively reducing the wave height in the shoaling and surf zone.

*4.4. Test 4: Rip Current*

In this subsection, the capacity of the proposed numerical model to simulate the wave propagation and wave-induced currents in a coastal area with a curvilinear coastline is verified. We numerically reproduced a laboratory test of a rip current test proposed by Hamm [24]. The laboratory test was carried out in a 30 × 30 m basin in which the curvilinear coastline was obtained by digging a rip channel along the central line of a 1:30 sloping beach. In this test, the incoming waves were monochromatic with a wave height of $H = 0.07$ m and a wave period of $T = 1.25$ s; the deep-water wavelength was $L = 3.5$ m. Since the laboratory basin was symmetric with respect to the alongshore coordinate $y$, in our numerical simulations only half of the basin was reproduced. On the symmetric axis of the laboratory basin (central line of the rip channel) and on the opposite lateral boundary, we imposed closed boundary conditions (null normal velocity and null gradient of the tangential velocity and free-surface elevation). On the bottom, a no-slip condition was imposed. The numerical simulations were carried out on a very coarse curvilinear grid consisting of 256 × 100 (25,600) computational cells in the horizontal plane: in the direction parallel to the deep-water wave fronts (y-axis), the average spatial grid step was about $\Delta y = 0.15$ m; in the wave propagation direction (x-axis), the average spatial grid step was about $\Delta x = 0.1$ m. The ratio between the number of grid cells along the wave propagation direction and the deep water wavelength was $r_L = 35$. The number of grid

points in the vertical direction was equal to five and the time step was equal to 0.001 s. The Smagorinsky coefficient $C_s$ was set to 0.1. The numerical simulations were carried out for 250 wave periods. The time average of the hydrodynamic quantities began after the first 50 wave periods and was carried over the next 200 wave periods. Figure 7a shows a plan view of the boundary-conforming curvilinear grid in which only one line in every two is drawn. Figure 7b,c show two vertical sections of the computational domain: Section A is located along the rip channel ($y_A = 14.96$ m) and Section B is located close to the opposite lateral boundary, where the bottom is a 1:30 sloping beach ($y_B = 1.98$ m). Figure 8 shows a three-dimensional view of the bottom.

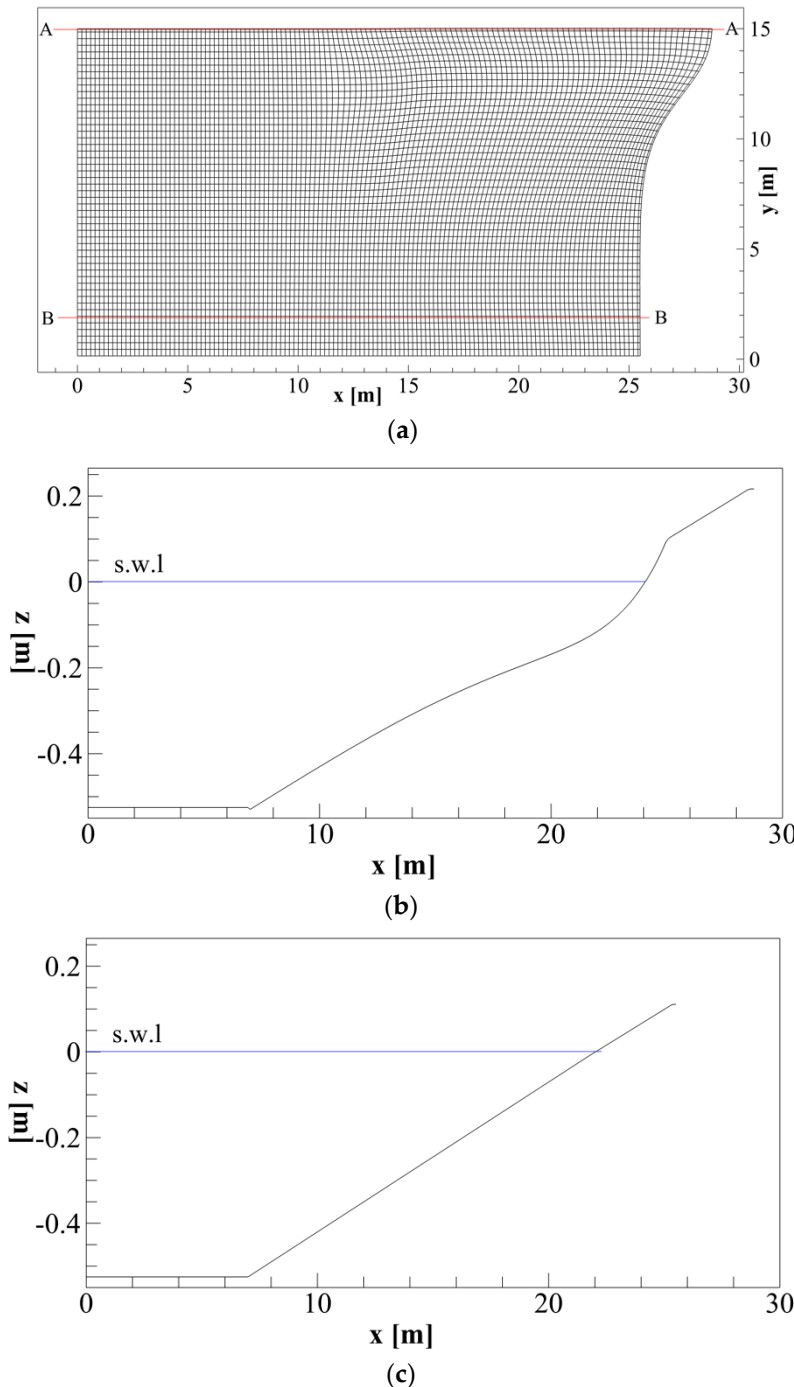

**Figure 7.** Test 4. Rip current. (**a**) computational grid (only one line in every 5 is drawn). (**b**) bottom profiles along Section A. (**c**) bottom profile along Section B.

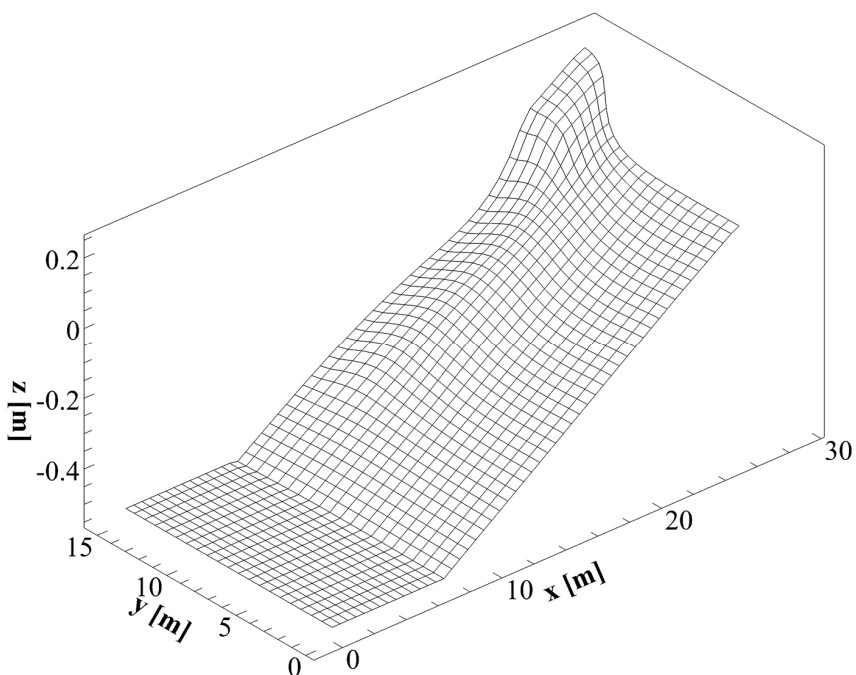

**Figure 8.** Test 4. Rip current. Three-dimensional view of the bottom.

This test was carried out using the proposed WTENO model, the 2TVD model proposed by Cannata et al. [3], and the 5WENO model. In Figure 9, an instantaneous wave field obtained with the WTENO model is shown. The presence of the rip channel on the lateral boundary of the computational domain induced a rip current. This offshore-directed current interacted with the incoming waves. As shown in Figure 9, the interaction between the rip current and the incoming waves produced an increase in the wave height and a curvature of the wave fronts close to the rip channel.

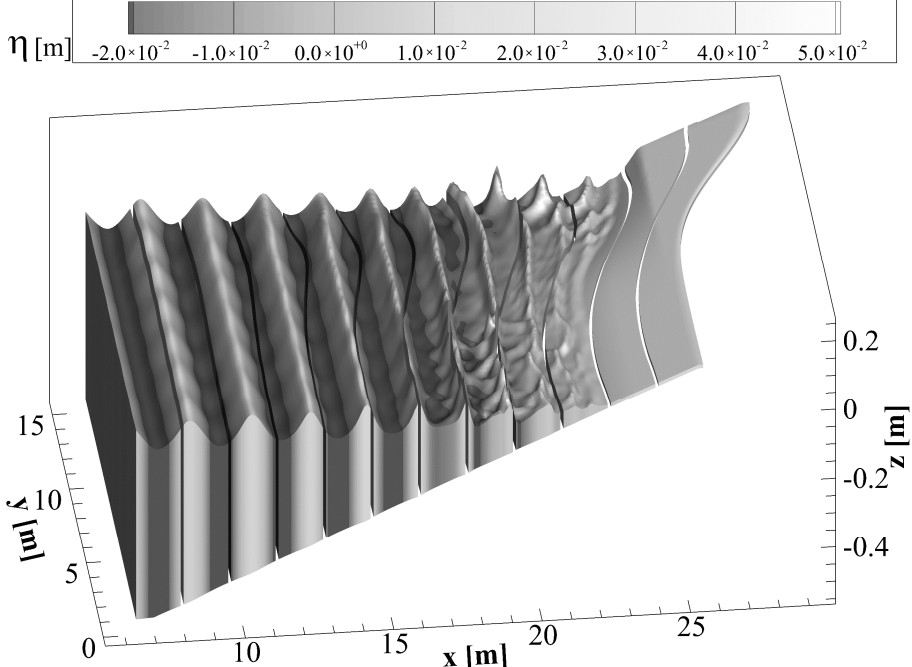

**Figure 9.** Test 4. Rip current. Instantaneous view of the wave field obtained using the proposed WTENO model.

In Figure 10a,b, the significant wave heights measured by Hamm [24] along the rip channel (Section A) and along the cross-shore section of the 1:30 sloping beach (Section B), respectively, are compared to the mean wave heights obtained using the WTENO model, the 2TVD model, and the 5WENO model. From Figure 10a,b it can be noted that, on this very coarse grid, the WTENO model provides results that are in good agreement with the experimental data: the wave evolution in the shoaling zone and the wave height reduction in the surf zone were well simulated; the wave breaking point was correctly located both in the rip channel and in the sloping beach. From the same figure, it can be noted that the numerical model based on second-order TVD reconstructions provided a wave height evolution that was highly inaccurate, in both the shoaling and the surf zone: the wave height increase in the incoming waves due to shoaling was completely absent and no wave breaking location could be found. From Figure 10 it can be seen that the results obtained using the 5WENO model are in better agreement with the experimental data than those obtained using the 2TVD model, but they were affected by similar drawbacks: the maximum wave height was overestimated and the wave breaking point was predicted too far offshore, especially along the cross-section of the 1:30 sloping beach.

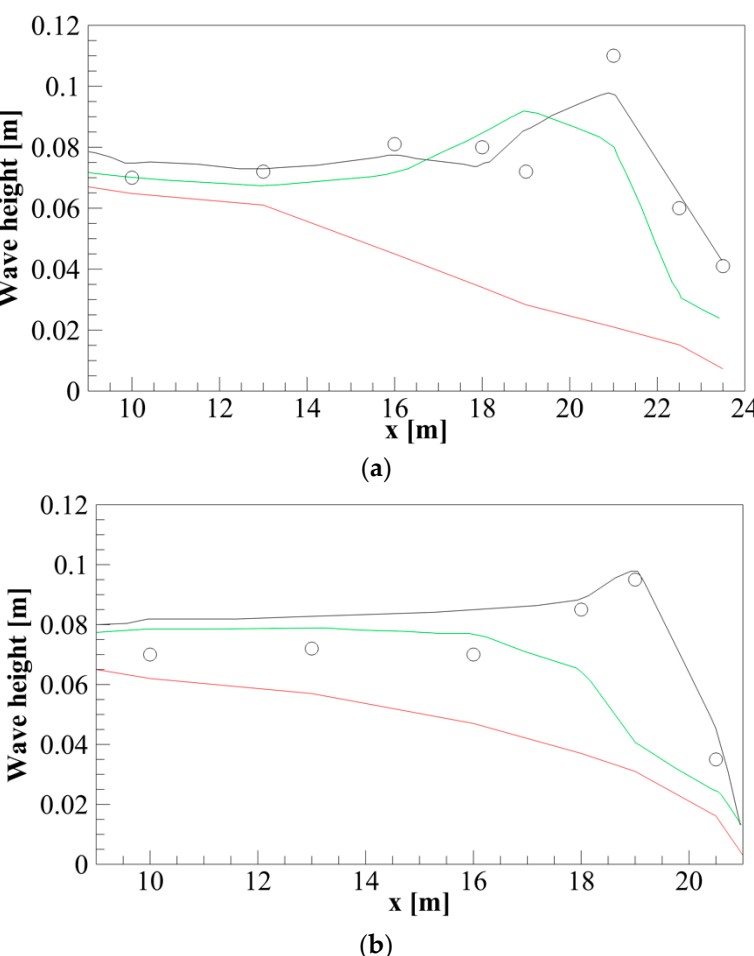

**Figure 10.** Test 4. Rip current. Wave height. Comparison between experimental and numerical results: significant wave height from Hamm [24] (circles); mean wave height obtained with the WTENO model (black line), the 2TVD model proposed by Cannata et al. [3] (red line), and the 5WENO model (green line). (**a**) Section A, along the rip channel; (**b**) Section B, along the cross-section of the 1:30 sloping beach.

Figure 11a,b (where only one vector in every three is drawn) show the induced wave current obtained using the WTENO model close to the free surface and close to the bottom, respectively. From Figure 11a it can be noted that, close to the free surface,

the alongshore component of the mean gradient of the free-surface elevation induced a significant alongshore current, which close to the channel became a rip current. This current produced a main anticlockwise circulation pattern and secondary clockwise recirculation structures close to the shoreline. From the same figure it can also be noted that close to the free surface, except in the rip channel, most of the cross-shore velocity components were directed onshore. On the contrary, from Figure 11b it can be seen that, close to the bottom, most of the cross-shore velocity components were directed offshore. These variations in the sign of the cross-shore current components along the vertical direction highlight the presence of undertow currents in the surf zone that make the circulation pattern fully three-dimensional.

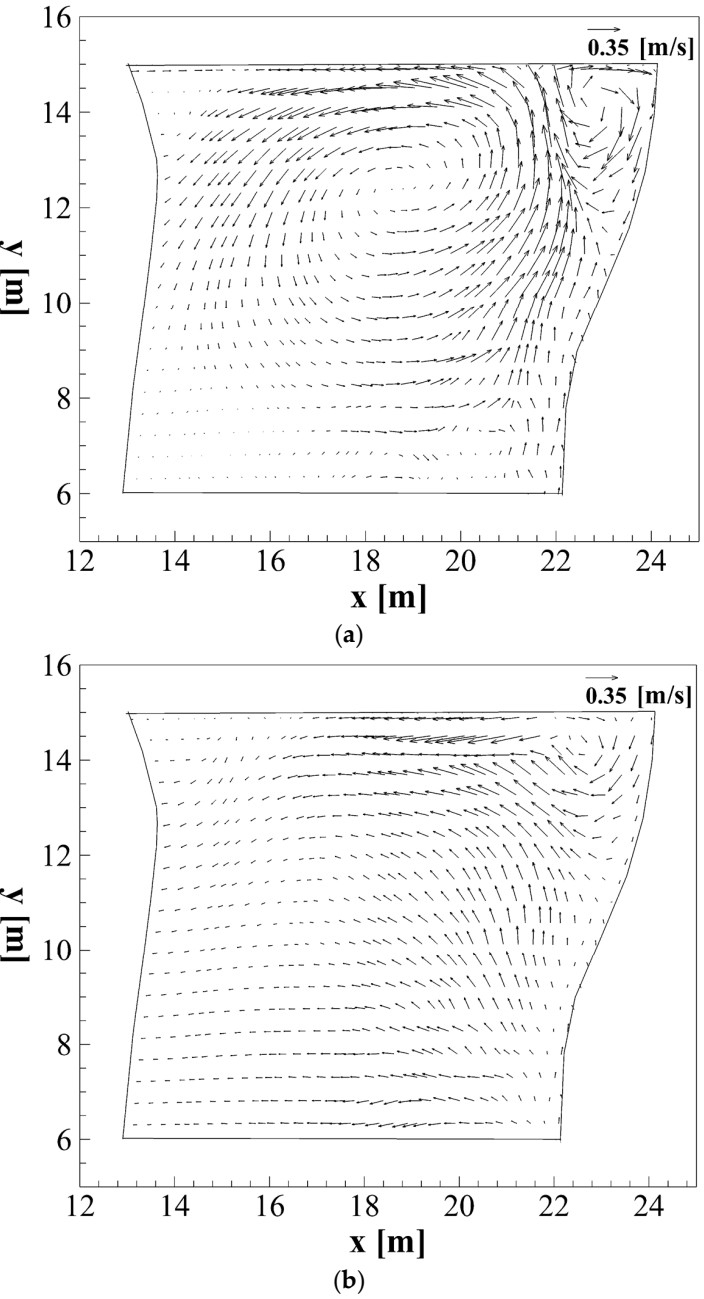

**Figure 11.** Test 4. Rip current. Plant view of the time average horizontal velocity components obtained using the proposed WTENO model: (**a**) close to the free surface; (**b**) close to the bottom (only one of every three vectors is drawn).

Figure 12 shows the near-bottom offshore-directed cross-shore current along the rip channel obtained using the WTENO model, in comparison with the currents obtained using the 2TVD model, the 5WENO model, and the experimental data from Hamm [24]. From this figure, it can be seen that the numerical results obtained with the proposed WTENO model are in good agreement with the experimental data. From the same figure it can be noted that, in this very coarse grid, the results obtained using the 2TVD model (red line) significantly underestimated the magnitude of the rip current. Such erroneous values of the offshore-directed cross-shore current component were due to a significant underestimation of the mean gradients of the free-surface elevation that drive the currents in the entire coastal basin. The main reason for these drawbacks is an excessive numerical energy dissipation introduced by the 2TVD model in the shoaling and surf zone which causes a wrong wave height evolution in the incoming waves (see Figure 10). Figure 12 shows that a significant underestimation of the rip current was also obtained using the 5WENO model (green line).

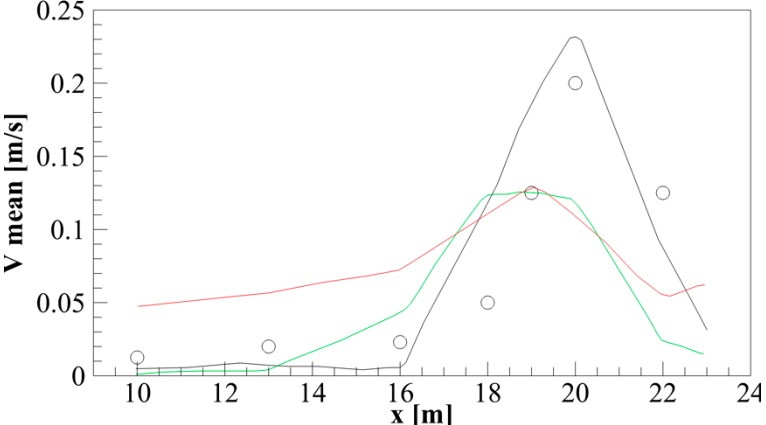

**Figure 12.** Test 4. Rip current. Time average of the near-bottom offshore-directed cross-shore velocity component along the rip channel. Experimental data from Hamm [24] (circles); WTENO model (black line); 2TVD model proposed by Cannata et al. [3] (red line); 5WENO model (green line).

The numerical results that are shown in this section highlight the capability of the proposed WTENO model to effectively simulate the wave height evolution in the shoaling zone, the maximum wave height, and the wave height reduction in the surf zone, also by using a very coarse computational grid in which the ratio $r_L$ (between the number of grid nodes in the wave propagation direction and the deep-water wavelength) is significantly less than 100. Consequently, on these coarse grids, differently from the numerical models based on second-order TVD and fifth-order WENO reconstructions, the proposed WTENO was able to simulate the spatial variations of the mean free-surface elevations and the wave-induced coastal currents.

### 4.5. Wave–Structure Interaction

In this subsection, we present a real application of the validated proposed model, consisting in the three-dimensional simulation of the interaction between breaking waves and an emerged breakwater. Emerged breakwaters are widely used coastal defense structures. Generally, they are heavy structures resting on the bottom which emerge from the free surface to avoid wave propagation. In most cases, they are placed parallel to the shoreline, isolated or in series. The presence of such barriers, besides modifying the wave fields and coastal currents, can cause local sea bottom erosion produced by quasi-periodic vortex structures close to the edge of the barrier. The flow velocity field that causes this local sea bottom erosion is fully three-dimensional and is related to the formation of vortices of various dimensions which interact with each other. In this subsection, the proposed model is applied to the numerical simulation of the complex free-surface elevation and

three-dimensional flow velocity fields produced by the interaction between breaking waves and an emerged vertical coastal barrier, which is placed parallel to the shoreline in the surf zone. We numerically reproduced a rectangular portion, 20 m × 6 m, of a coastal area in which an emerged parallelepiped barrier (2 m long, 0.5 m wide, and 1 m high) was placed parallel to the shoreline. In this coastal area, an initial undisturbed water depth of $h = 0.4$ m (between $x = -5$ m and $x = 0$ m) was followed by a 1 : 35 sloping beach. In Figure 13a,b, a plan view and a vertical section, respectively, of the coastal area with the coastal barrier are shown.

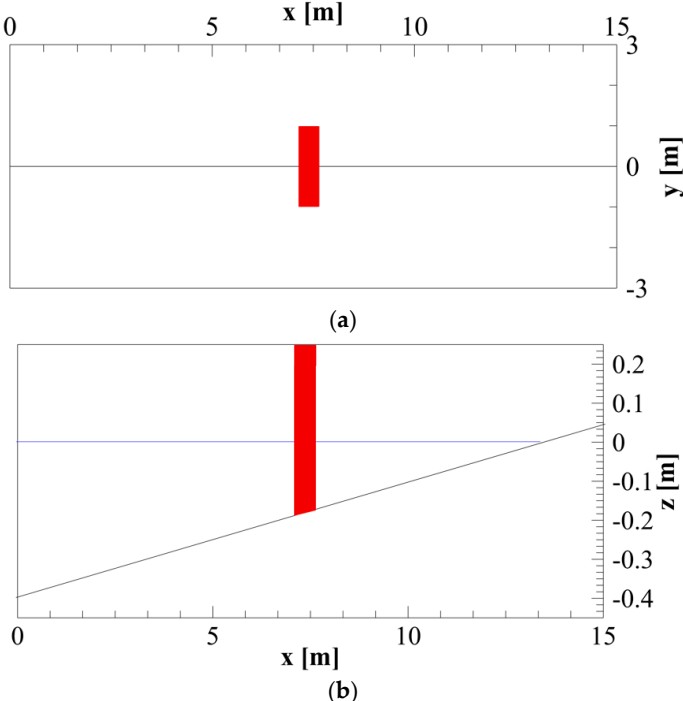

**Figure 13.** Wave–structure interaction. Coastal area with a vertical emerged barrier (in red). (**a**) Plan view. (**b**) Vertical section.

As can be seen from Figure 13a, the central line parallel to the x-axis is an axis of symmetry of the above-mentioned coastal area. Consequently, to save computational time, we numerically reproduced only one half of the original area. In Figure 14, the computational grid adopted for the above-mentioned numerical simulation is shown (in which only one line in every three is drawn): the waves were generated at $x = -5$ m and propagated along the x-axis in the direction of increasing $x$; the barrier was located between $x = 7.1$ m and $x = 7.6$ m, and between $y = 0$ m and $y = 1$ m. In the horizontal directions, the spatial discretization steps ranged from $\Delta x = 0.025$ m to $\Delta x = 0.05$ m and from $\Delta y = 0.025$ m to $\Delta y = 0.05$ m. In the vertical direction, we adopted 13 non-uniformly distributed grid cells.

On the right (with respect to the wave propagation direction) lateral boundary ($y = 0$ m), we imposed a closed boundary condition (null normal gradient of the free-surface and tangential flow velocity components and zero normal velocity). On the left lateral boundary ($y = 3$ m), we imposed an open boundary condition (null normal gradient of free-surface elevation and flow velocity components). On the left boundary, at x = 0 m, a cnoidal wave with a period of $T = 2$ s and a wave height of $H = 0.125$ m was imposed. The Smagorinsky coefficient was set $C_s = 0.2$. Figure 15 shows a sequence of instantaneous snapshots of the free surface obtained using the proposed model during the numerical simulation of the wave–structure interaction (the barrier was drawn as transparent for purposes of clarity).

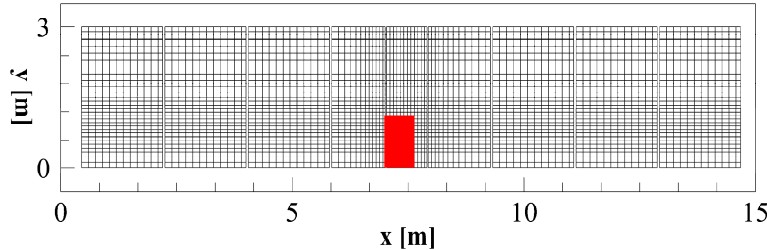

**Figure 14.** Wave–structure interaction. Plan view of the computational grid with the vertical emerged barrier (in red).

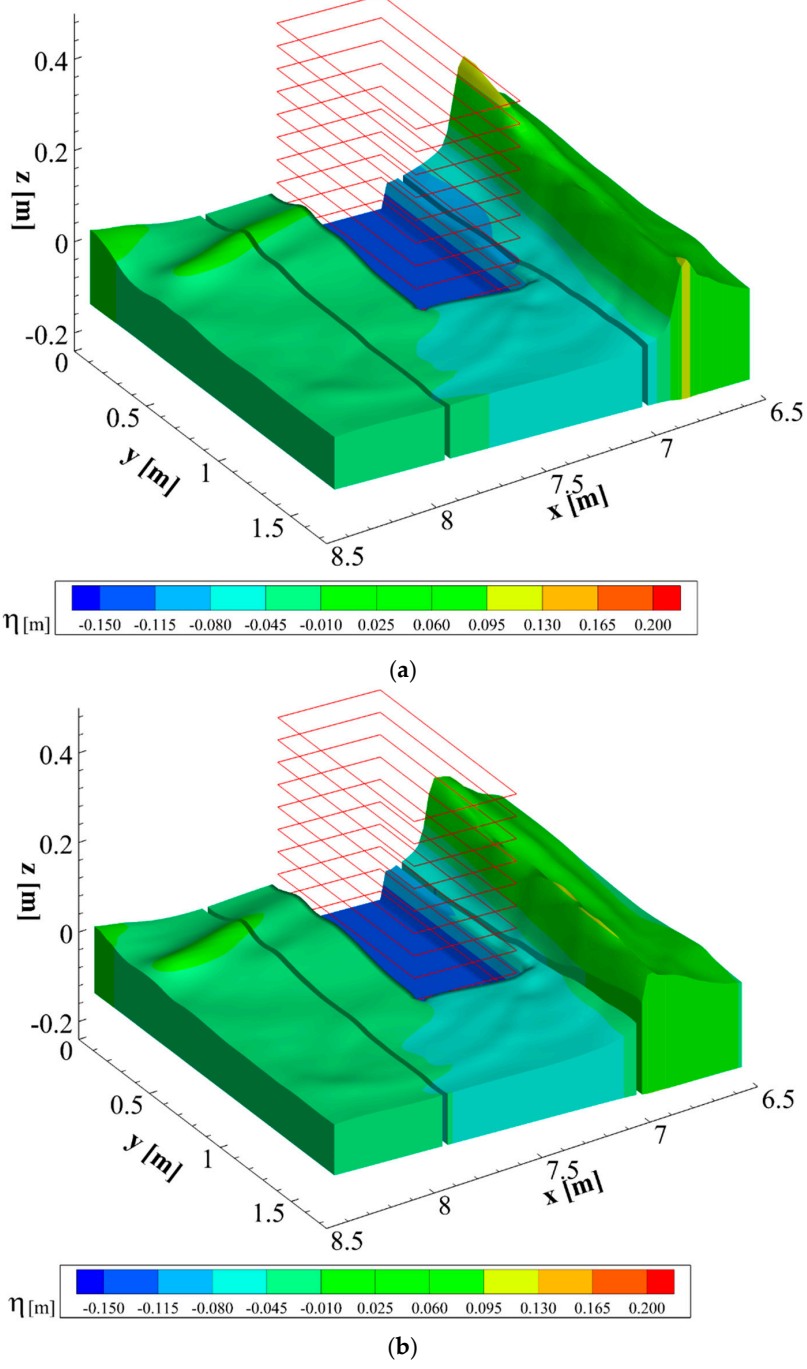

**Figure 15.** *Cont.*

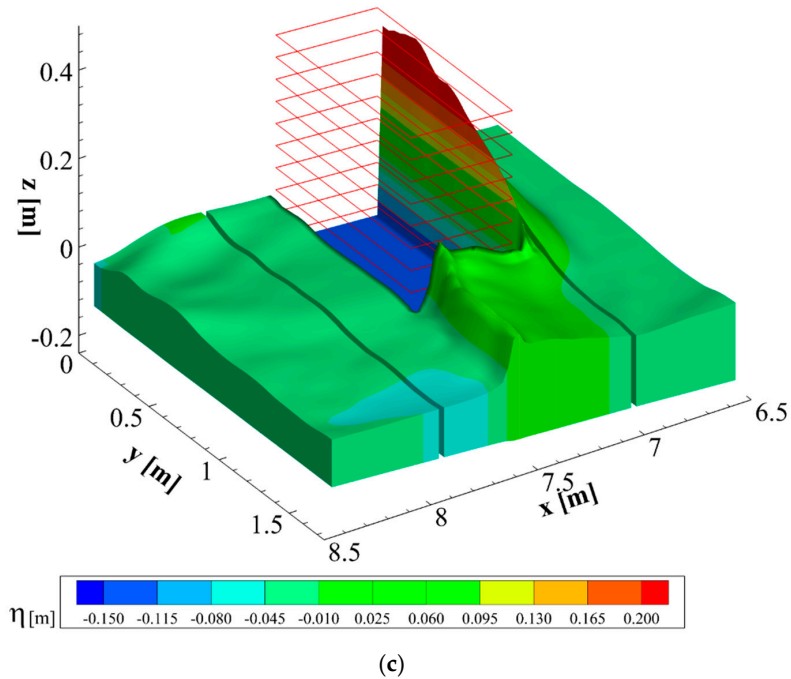

(**c**)

**Figure 15.** Wave–structure interaction. Instantaneous snapshots of the free surface obtained using the proposed model (the barrier was drawn as transparent for purposes of clarity). Three different instantaneous wave field: (**a**) $T = 100$ s, (**b**) $T = 101$ s and (**c**) $T = 102$ s.

From Figure 15a,c it is possible to see the reduction in the wave height due to the wave breaking (before reaching the barrier), a significant increase in the surface elevation due to the impact with the barrier, and the propagation of the breaking front wave along the side of the barrier. The spatial variations in the free-surface elevation (due to the presence of the barrier) produced significant modifications in the flow velocity fields and characteristic wave-induced circulation patterns.

Figure 16 shows the circulation patterns obtained by averaging over time the flow velocity calculated with the proposed model at different vertical distances from the bottom: (a) near the bottom; (b) at an intermediate water depth; (c) near the free surface (only one out of every two vectors are drawn). As can be seen from Figure 16, upstream of the barrier, the wave trains which impacted the barrier induced a mean gradient of the free-surface elevation driving flow velocities parallel to the barrier and directed toward the zone where the barrier is absent. Such a velocity component was greater close to the bottom and gradually decreased upon approaching the free surface. Downstream of the barrier, a mean gradient of the free-surface elevation induced flow velocities that were parallel to the barrier and directed towards the sheltered zone. Consequently, behind the barrier, there was a clockwise vortex with increasing intensity upon approaching the free surface.

Figure 17 shows the instantaneous local vortex structures close to the edges of the barrier and propagating toward the shoreline. These vortex structures were visualized using the Q-criterion, which is a vortex identification method based on the three-dimensional contours of the second invariant of the velocity gradient tensor. As can be seen from Figure 17, the presence of the barrier caused, near the edges, the onset of quasi-periodic vortex structures with a vertical axis. Such vortex structures emerged close to the bottom and developed toward the free surface. Downstream of the barrier, the vortices generated at the upstream edge of the barrier were characterized by almost horizontal axes, which were directed toward the shoreline. Such vortices were stretched by the wave motion and breakdown to form smaller vortex structures.

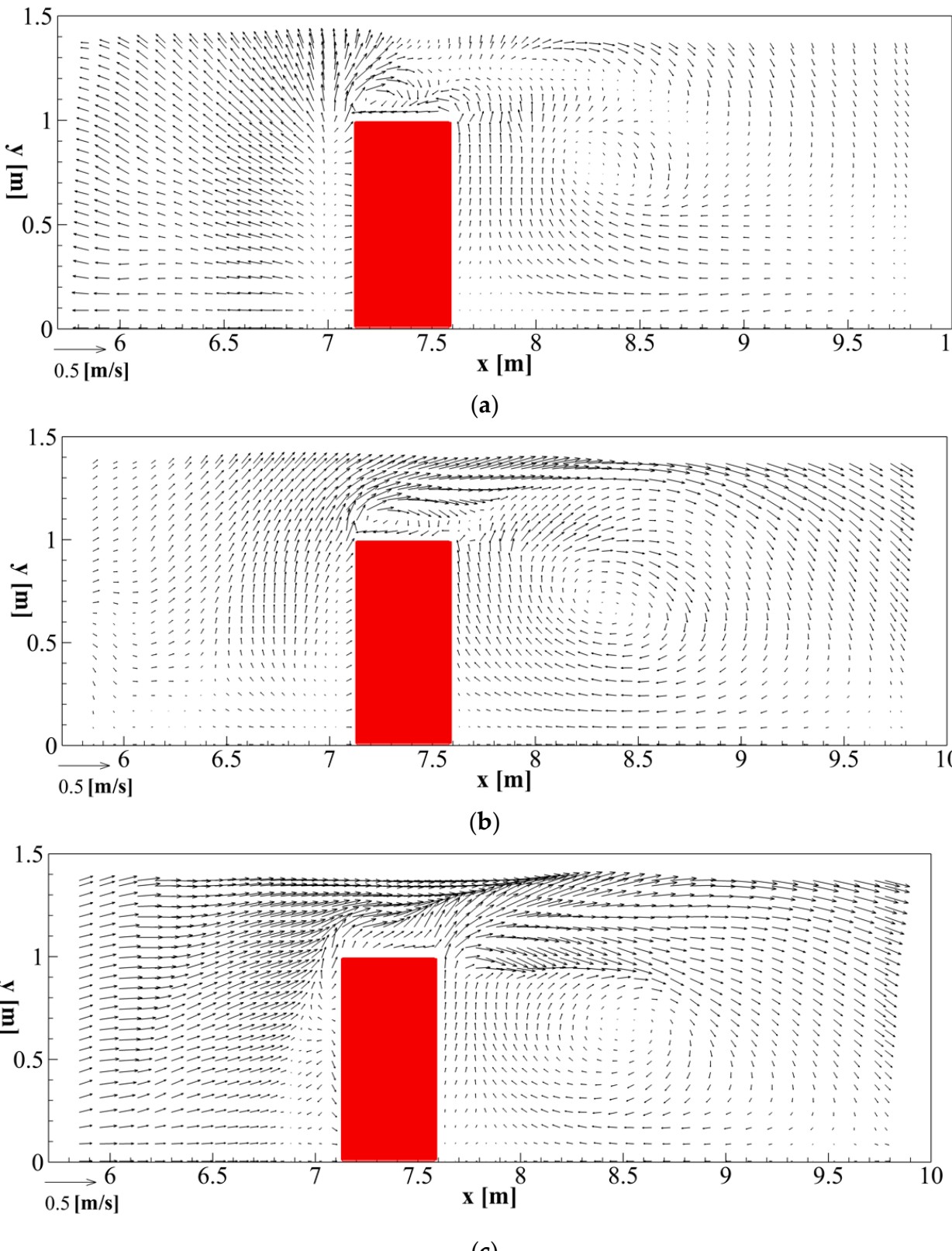

**Figure 16.** Wave–structure interaction. Circulation patterns. (**a**) Near the bottom. (**b**) At an intermediate water depth. (**c**) Near the free surface.

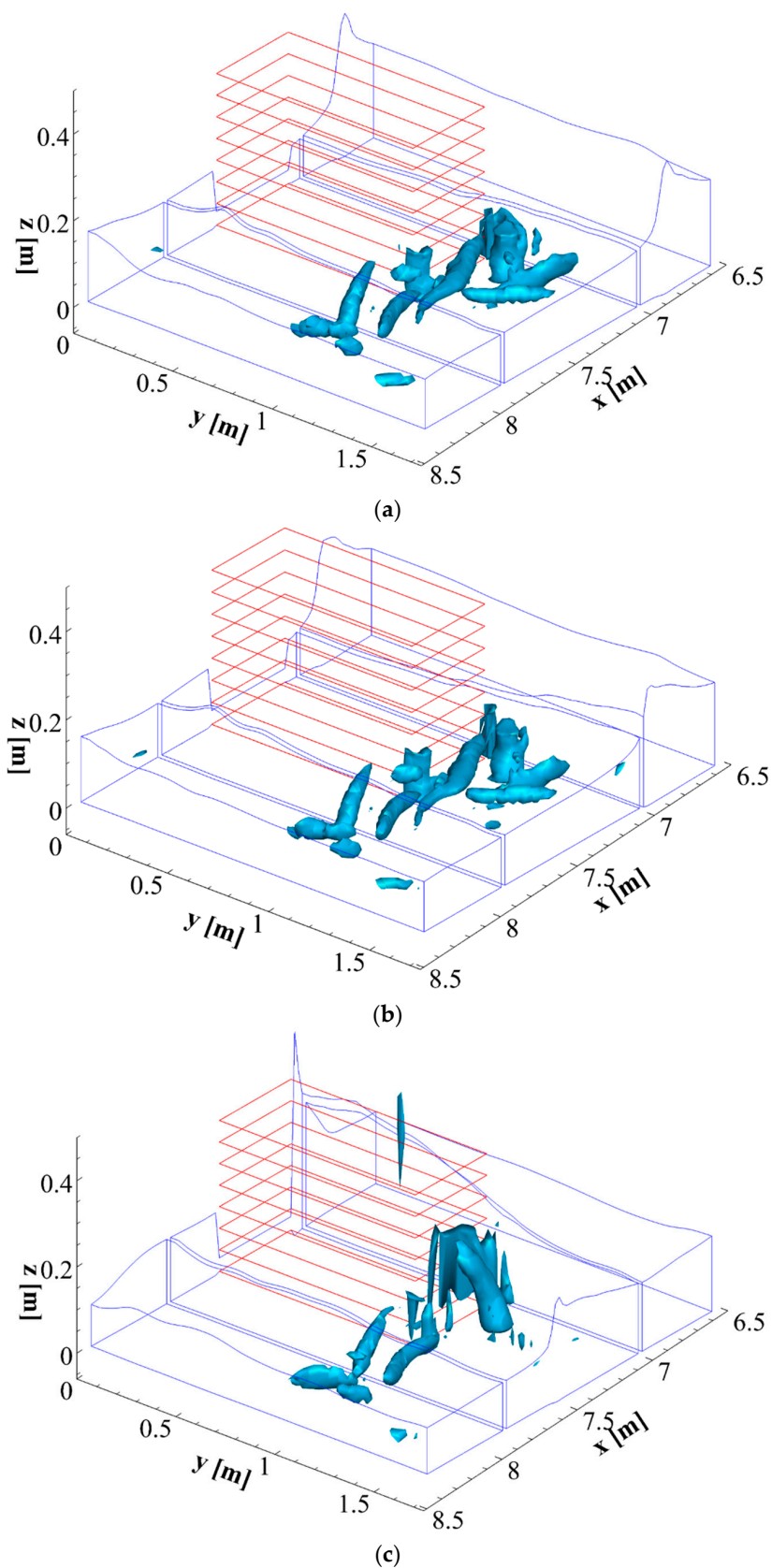

**Figure 17.** Wave–structure interaction. Instantaneous local vortex structures visualized using the Q-criterion (three-dimensional contours of the second invariant of the velocity gradient tensor). Three different instantaneous wave field: (**a**) $T = 100$ s, (**b**) $T = 101$ s and (**c**) $T = 102$ s.

The numerical results shown in this subsection highlight the ability of the proposed model to simulate the complex fully three-dimensional hydrodynamic phenomena produced by the interaction between breaking waves and an emerged barrier. Especially noteworthy are the formation of large-scale circulation patterns and the emergence of local quasi-periodic vortex structures near the structure, at the sea bottom. Such local fully three-dimensional hydrodynamic phenomena can cause an increment in the capacity of the wave motion to put in suspension and carry away the suspended sediment from the bottom, with a possible partial undermining of the barrier.

## 5. Conclusions

A new three-dimensional high-order shock-capturing model for the numerical simulation of breaking waves was proposed. The proposed model is based on the integral contravariant form of the Navier–Stokes equations in a time-dependent generalized curvilinear coordinate system proposed by [3]. Such an integral contravariant form of the equations of motion is numerically integrated by a new conservative numerical scheme, in which exclusively the conserved variables are used. In the proposed numerical scheme, the point values of the conserved variables on the cell faces of the computational cells were obtained using an original reconstruction procedure called WTENO, designed for the 3D numerical simulation of breaking waves. On such cell faces, the time evolution in the discontinuity was calculated using the exact solution of the Riemann problem. The proposed model was validated by numerically reproducing several experimental tests of breaking waves on computational grids that were significantly coarser than those usually used in the literature to validate the existing 3D shock-capturing models. The results obtained using the proposed model were also compared with those obtained using the model proposed by Cannata et al. [3], which is based on second-order TVD reconstructions and an approximate Riemann solver usually adopted in the literature. The above comparison shows that the results obtained with the 3D shock-capturing model based on second-order TVD reconstructions and approximate Riemann solvers were affected by some main drawbacks: the wave height evolution in the shoaling zone and the maximum wave height were underestimated and the predicted wave breaking point was incorrectly located. The same comparison showed that the proposed model was able to overcome the above-mentioned drawbacks: it correctly simulated the wave height increase in the shoaling zone, and correctly predicted the location of the wave breaking point, the maximum wave height, and the wave height decay in the surf zone. The validated proposed model was applied to a real case consisting in the simulation of the complex three-dimensional flow fields and free-surface elevation generated by the interaction between breaking waves and an emerged barrier. The numerical results show the ability of the proposed model to simulate both large-scale circulation patterns downstream of the barrier and the onset of quasi-periodic vortex structures close to the edge of the barrier. These fully three-dimensional complex flow structures can put in suspension and carry away the suspended sediment from the bottom and cause local scour near the barrier.

**Author Contributions:** Conceptualization, F.G. and G.C.; methodology and validation, F.P. and B.I. All authors have read and agreed to the published version of the manuscript.

**Funding:** This research received no external funding.

**Institutional Review Board Statement:** Not applicable.

**Informed Consent Statement:** Informed consent was obtained from all subjects involved in the study.

**Data Availability Statement:** Not applicable.

**Conflicts of Interest:** The authors declare no conflict of interest.

## Appendix A

The $x^1$-split Riemann problem is defined as an initial value problem

$$
\begin{cases}
\dfrac{\partial U}{\partial t} + \dfrac{\partial F(U)}{\partial x^1} = 0 \\
U(x,0) = \begin{cases} U_L \ if \ x^1 < 0 \\ U_R \ if \ x^1 > 0 \end{cases}
\end{cases}
\tag{A1}
$$

where

$$
U_L = \begin{pmatrix} H_L \\ H_L u_L \\ H_L v_L \\ H_L w_L \end{pmatrix}; \quad
U_R = \begin{pmatrix} H_R \\ H_R u_R \\ H_R v_R \\ H_R w_R \end{pmatrix}; \quad
F(U) = \begin{pmatrix} Hu \\ Huu + 0.5gH^2 \\ Huv \\ Huw \end{pmatrix}
\tag{A2}
$$

$F(U)$ is the vector of fluxes in the $x^1$-direction; $U_L$ and $U_R$ are the vectors of the initial conditions on the left and right of the cell face, which in the local system of coordinates is placed at $x = 0$. The structure of the solution of the Riemann problem is given by three waves, associated with the eigenvalues $e1 = u - a$, $e2 = u$, and $e3 = u + a$ (in which $a = \sqrt{gh}$), which separate four constant states, indicated by $W_L$, $W_{stL}$, $W_{stR}$, and $W_R$, where $W = (H, u, v, w)$. The values $W_{stL}$ and $W_{stR}$ denote the star region, where the solution is unknown, and arise from the interaction of $W_L$ and $W_R$. Across the left and right waves, $H$ and $u$ change, while $v$ and $w$ remain constant; across the middle wave, $v$ and $w$ change discontinuously, while $H$ and $u$ remain constant. We denote by $H_{st}$ and $u_{st}$ the constant values of the water depth and x-component velocity in the star region, respectively. The left and right waves are determined according to the following conditions:

$$
H_{st} > H_L : \ left \ wave \ is \ a \ shock \ wave
$$
$$
H_{st} \leq H_L : \ left \ wave \ is \ a \ rarefaction \ wave
$$

and

$$
H_{st} > H_R : \ right \ wave \ is \ a \ shock \ wave
$$
$$
H_{st} \leq H_R : \ right \ wave \ is \ a \ rarefaction \ wave
$$

The tangential velocity components $v$ and $w$ do not influence the left and right waves and the values of $H_{st}$ and $u_{st}$ and, therefore, are not taken into account in the solution procedure. The first step of the procedure consists in obtaining a non-linear algebraic equation for $H_{st}$. For this purpose, we relate $u_*$ to the left and right values of $H$ by functions $fun_L(H, H_L)$ and $fun_R(H, H_R)$, which govern the relations between $u$ and $H$ across the left and right waves, respectively. Using the Riemann invariants for a rarefaction wave and the Rankine–Hugoniot conditions for a shock wave, we obtain

$$
fun_L = \begin{cases} 2\left(\sqrt{gH_{st}} - \sqrt{gH_L}\right), \ if \ H_{st} \leq H_L \ (rarefaction \ wave) \\ (H_{st} - H_L)\sqrt{\dfrac{1}{2}g\dfrac{(H_{st}+H_L)}{H_{st}H_L}}, \ if \ H_{st} > H_R \ (shock \ wave) \end{cases}
$$
$$
fun_R = \begin{cases} 2\left(\sqrt{gH_{st}} - \sqrt{gH_R}\right), \ if \ H_{st} \leq H_R \ (rarefaction \ wave) \\ (H_{st} - H_R)\sqrt{\dfrac{1}{2}g\dfrac{(H_{st}+H_R)}{H_{st}H_R}}, \ if \ H_{st} > H_R \ (shock \ wave) \end{cases}
\tag{A3}
$$

The solution $H_*$ is given by the root of the algebraic equation

$$
fun(H_{st}) \equiv fun_L(H_{st}, H_L) + fun_R(H_{st}, H_R) + u_R - u_L = 0;
\tag{A4}
$$

Three intervals of values of $H_{st}$ in which the solution of the Riemann problem is determined are of physical interest ($H_{st} > 0$):
$H_{st} < H_{min}$; $H_{min} \leq H_{st} \leq H_{Max}$; $H_{st} > H_{Max}$ where $H_{min} = min(H_L, H_R)$ and $H_{Max} = max(H_L, H_R)$.

Given $H_L$ and $H_R$, the solution of Equation (A4) depends exclusively on $\Delta u = u_R - u_L$. In order to obtain positive values of $H_{st}$, the following depth positivity condition, $2(a_L + a_R) > u_R - u_L$, must be fulfilled.

In the case in which $f(H_{min}) > 0$, the left and right waves are rarefaction waves and $fun(H_{st})$ becomes

$$2(a_{st} - a_L) + 2(a_{st} - a_R) + u_R - u_L = 0 \tag{A5}$$

The solutions for $a_{st}$ (and thus for $H_{st}$) and for $u_{st}$ are given by

$$a_{st} = \frac{1}{2}(a_L + a_R) - \frac{1}{4}(u_R - u_L); \; u_{st} = \frac{1}{2}(u_L + u_R) + a_L - a_R \tag{A6}$$

In all the other cases, Equation (A4) for $H_*$ is numerically solved with a Newton–Raphson iterative scheme. Once $H_{st}$ is known, the value of $u_{st}$ is calculated using the following equation:

$$u_{st} = \frac{1}{2}(u_L + u_R) + \frac{1}{2}[f_R(H_{st}, H_R) - f_L(H_{st}, H_L)] \tag{A7}$$

If $H_{st} > H_L$, the left wave is a shock wave with speed $S_L$ given by

$$S_L = u_L - a_L q_L$$
$$q_L = \sqrt{\frac{\frac{1}{2}(H_{st} + H_L)H_{st}}{H_L^2}} \tag{A8}$$

and the structure of the complete solution is shown in Figure A1a.

If $H_{st} \leq H_L$, the left wave is a rarefaction wave, for which the speeds of the head $S_{HL}$ and tail $S_{TL}$ are

$$S_{HL} = u_L - a_L$$
$$S_{TL} = u_{st} - a_{st} \tag{A9}$$

In this case, the solution inside the rarefaction wave is given by

$$W_{Lfan} = \begin{cases} a = \frac{1}{3}\left(u_L + 2a_L - \frac{x^1}{t}\right) \\ u = \frac{1}{3}\left(u_L + 2a_L + \frac{2x^1}{t}\right) \end{cases} \tag{A10}$$

and the structure of the complete solution is shown in Figure A1b.

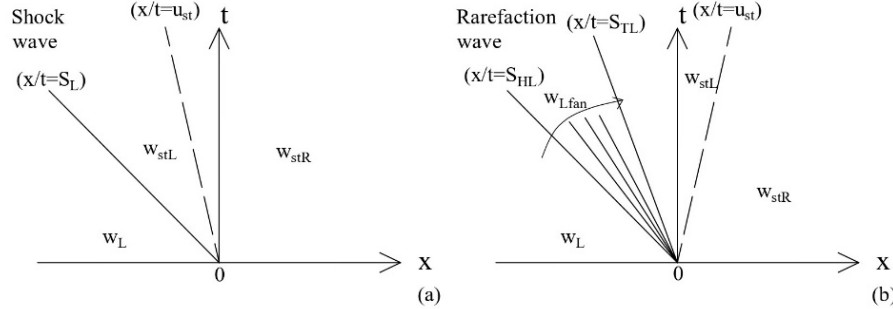

**Figure A1.** Solution of the Riemann problem to the left of the contact wave: (**a**) left wave is a shock wave; (**b**) left wave is a rarefaction wave.

If $H_{st} > H_R$, the right wave is a shock wave, with speed $S_R$ given by

$$\begin{cases} S_R = u_R + a_R q_R \\ q_R = \sqrt{\frac{\frac{1}{2}(H_{st} + H_R)H_{st}}{H_R^2}} \end{cases} \tag{A11}$$

and the structure of the complete solution is shown in Figure A2a.

If $H_{st} \leq H_R$, the right wave is a rarefaction wave, for which the speeds of the head $S_{HR}$ and tail $S_{TR}$ are

$$
\begin{aligned}
S_{HR} &= u_R + a_R \\
S_{TR} &= u_{st} + a_{st}
\end{aligned}
\tag{A12}
$$

In this case, the solution inside the right rarefaction wave is given by

$$
\begin{aligned}
S_{HR} &= u_R + a_R \\
S_{TR} &= u_{st} + a_{st}
\end{aligned}
W_{Rfan} =
\begin{cases}
a = \frac{1}{3}\left(-u_R + 2a_R + \frac{x^1}{t}\right) \\
u = \frac{1}{3}\left(u_R - 2a_R + 2\frac{x^1}{t}\right)
\end{cases}
\tag{A13}
$$

and the structure of the complete solution is shown in Figure A2b.

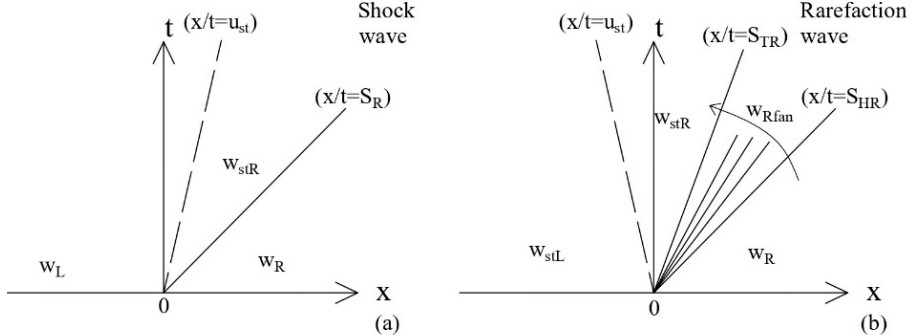

**Figure A2.** Solution of the Riemann problem to the right of the contact wave: (**a**) right wave is a shock wave; (**b**) right wave is a rarefaction wave.

Once $H_{st}$ and $u_{st}$ are calculated, the remaining velocity components are easily obtained as a function of the celerity of the contact wave:

$$
\text{if } u_{st} > 0, v = v_L \text{ and } w = w_L; \quad \text{if } u_{st} < 0, v = v_R \text{ and } w = w_R
\tag{A14}
$$

**Appendix B**

**Table A1.** Symbols used in this paper.

| Symbols | Unit of Measurements | |
|---|---|---|
| $\Delta V(t)$ | (m$^3$) | Moving control volume |
| $\Delta A(t)$ | (m$^2$) | Boundary surface of the moving control volume |
| $x^l$ $(l = 1, 3)$ | (m) | Cartesian coordinates |
| $\vec{u}$ | (m s$^{-1}$) | Cartesian flow velocity vector |
| $u, v, w$ | (m s$^{-1}$) | Cartesian components of flow velocity vector |
| $\vec{\omega}$ | (s) | Cartesian velocity vector of the boundary surface control volume |
| $\vec{n}$ | (-) | Cartesian outward normal unit vector |
| $\cdot$ | (-) | Dot product |
| $:$ | (-) | Doubly contracted dot product |
| $\otimes$ | (-) | Tensor product between vectors |
| $\vec{\lambda}$ | (-) | Generic Cartesian vector |
| $G$ | (m s$^{-2}$) | Acceleration due to gravity |
| $\eta$ | (m) | Free-surface elevation |
| $H$ | (m) | Total water depth |
| $h$ | (m) | Still water depth |
| $\rho$ | (kg m$^{-3}$) | Fluid density |
| $p$ | (kg s$^{-2}$ m$^{-1}$) | Dynamic pressure |
| $P$ | (kg s$^{-2}$ m$^{-1}$) | Total pressure |
| $\underline{R}$ | (kg s$^{-2}$ m$^{-1}$) | Stress tensor |
| $\underline{S}$ | (s$^{-1}$) | Stain rate tensor |
| $\underline{I}$ | (s$^{-1}$) | Identity tensor |
| $\xi^l$ $(l = 1, 3)$ | (-) | Curvilinear coordinates |
| $\vec{g}_{(l)}$ $(l = 1, 3)$ | (m) | *l-th* covariant base vector |



**Table A1.** *Cont.*

| Symbols | Unit of Measurements | |
|---|---|---|
| $\vec{g}^{(l)}\,(l=1,3)$ | (m$^{-1}$) | *l-th* contravariant base vector |
| $\sqrt{g}$ | (m$^3$) | Jacobian of the transformation |
| $\wedge$ | (-) | Symbol for the vector product |
| $\sqrt{g0}$ | (m$^2$) | Factor of the Jacobian of the transformation |
| $\vec{k}$ | (-) | Cartesian vertical unit vector |
| $\vec{\tilde{g}}^{(l)}\,(l=1,3)$ | (m$^{-1}$) | Contravariant base vector at the center of the control volume |
| $\lambda_m\,(m=1,3)$ | (-) | $m$thcovariant component of $\vec{\tilde{g}}^{(l)}$ |
| $u^l\,(l=1,3)$ | (m s$^{-1}$) | *l-th* Contravariant component of the flow velocity vector |
| $\Delta\xi^1\Delta\xi^2\Delta\xi^3$ | (-) | Control volume in the transformed space |
| $\Delta A_0^{\alpha+},\,\Delta A_0^{\alpha+}\,(\alpha=1,3)$ | (-) | Boundary surfaces of the control volume on which the coordinate $\xi^\alpha$ is constant |
| $R^{m\alpha}\,(m,\alpha=1,3)$ | (s$^{-2}$) | Contravariant components of the stress tensor without the pressure term |
| $\omega^\alpha\,(\alpha=1,3)$ | (s$^{-1}$) | $\alpha$-*th* Contravariant component of the velocity of the moving coordinate |
| $I_{i,j,k}$ | (-) | Generic hexahedral computational cell |
| $(\,)$ | | Cell-averaged value |
| $(\tilde{\,})$ | | Surface-averaged value |
| $(\hat{\,})$ | | Line-averaged value |
| $(\,)^+,\,(\,)^-$ | | Point values on the cell face |
| $(\,)^{RS}$ | | Updated point value obtained with the Riemann solver |
| $(\,)^*$ | | Predictor value of a cell average variable |
| $\Phi$ | (m$^2$ s$^{-1}$) | Scalar potential |
| $g^{ls}(l,s=1,3)$ | (m$^{-2}$) | Contravariant metric tensor |
| $\Omega_p$ | (-) | Non-linear weights |
| $\delta_p$ | (-) | Cut-off functions |
| $c_p$ | (-) | Linear weight |
| $\Gamma_p$ | (-) | Normalized regularity function |
| $C_T$ | (-) | Dynamic threshold |
| $\gamma_p$ | (-) | Regularity function |
| $\beta_p$ | (-) | Smoothness indicator |
| $\tau_p$ | (-) | Global smoothness indicator |
| $C,\mu,\epsilon,d,B_l,B_h$ | (-) | WTENO coefficients |
| $n$ | (-) | Exponent of the dynamic threshold |
| $\theta$ and $\theta_2$ | (-) | WTENO functions |
| $\Psi$ | (m s$^{-1}$) | Threshold for local time rate of change |
| $\mathbf{U}$ | | Vector of the Cartesian conserved variables |
| $\mathbf{F(U),\,G(U),\,H(U)}$ | | Flux vectors |
| $\mathbf{S}$ | | Source term vector |
| $\nu_t$ | (m$^2$ s$^{-1}$) | Eddy viscosity |
| $C_s$ | (-) | Smagorinsky coefficient |
| $\Delta$ | (m) | Filter width |
| $(\,)_R,\,(\,)_L$ | | Right and left values on the cell face |
| $e1,\,e2,\,e3$ | (m s$^{-1}$) | Eigenvalues |
| $a$ | (m s$^{-1}$) | Celerity |
| $(\,)_{st}$ | (-) | Star region values |
| $S_L,\,S_R$ | (m s$^{-1}$) | Left and right wave speed |
| $S_H,\,S_T$ | (m s$^{-1}$) | Head and tail wave speed |

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
