# Peer review of "A Wave-Targeted Essentially Non-Oscillatory 3D Shock-Capturing Scheme for Breaking Wave Simulation"

_jmse, doi:10.3390/jmse10060810_

Round 1

Reviewer 1 Report

 A new three-dimensional high-order shock-capturing model for the numerical simulation of breaking waves is proposed. The proposed model is based on an integral contravariant form of the Navier-Stokes equations in a time-dependent generalized curvilinear coordinate system. The manuscript is not well organized. The authors should focus to revise the manuscript and write it like a research paper not a chapter of a book.

1.       The introduction section should be improved avoiding lump sum references such as [11-14,19-28]; all references should be cited with detailed and specific descriptions.  Include novelty.

2.       What is the most important physical conclusion making the work publishable in this journal? Mention this physical outcome in the Abstract.

3.       Nomenclature should be added to represents all mathematical symbols with units

4.       Lines 160 ,233,503, Error! Reference source not found it is not clear

5.       Many section are overlapping with published works special in introduction section , reduce the similarity rate

  1. Formulation needs improvement, a carful revision required .

7.       Some symbols are misplaces in some equations, carefully revise and correct

  1. The authors Should make comparison with previously published results.
  2. Should include a section to discuss the numerical method used in this study.
  3. The discussion should include examples of some real applications.

11.   Proper validation of the model is needed also supported by past studies

12.   Discussion section should be improved, It should not be just increase/decrease include some real discussion, the discussion should include examples of some real applications.

13.   Why such method used, include details

14.   Figures quality are very low

15.   Serval graphs show same effect but the parameter has different behavior why?  Figures quality needs improvement.

Reviewer 2 Report

The paper develops a new high-order shock-capturing model, which can well reproduce wave breaking. The organization and the English language are appropriate. It can be accepted by the journal with considering the following suggestion:

In Section 2, the derivation of the mathematical formulation is abstract. The transformation from the classic expression of Navier-Stokes equations to the curvilinear coordinate should be present in detail. The integral contravariant form of the momentum balance equation and continuity equation should be expressed in the discrete equation rather than the integral form, as the numerical model is based on the discrete equation of each vertical layer.

Round 2

Reviewer 1 Report

The manuscript is well revised, I recommend to be published.